# Interactions between ocean alkalinity enhancement and phytoplankton in an Earth System Model

Miriam Seifert<sup>1</sup>, Christopher Danek<sup>1</sup>, Christoph Völker<sup>1</sup>, and Judith Hauck<sup>1,2</sup>

<sup>1</sup>Alfred-Wegener-Institut Helmholtz-Zentrum für Polar- und Meeresforschung, 27570 Bremerhaven, Germany

<sup>2</sup>Universität Bremen, 28359 Bremen, Germany

**Correspondence:** Miriam Seifert (miriam.seifert@awi.de)

Abstract. The functioning and efficiency of ocean alkalinity enhancement (OAE) as a  $CO_2$  removal strategy is well investigated in model studies, but risks for the ecosystem are presently not considered in models. Our study examines OAE-phytoplankton feedbacks in an Earth System Model by adding carbonate system dependencies to the phytoplankton growth term. OAE is performed between 2040 and 2100 in the exclusive economic zones of Europe, the US, and China, with alkalinity additions reaching 103.2 Tmol year<sup>-1</sup> by the end of the century. Atmospheric  $pCO_2$  is reduced by  $3-8 \mu$ atm. The excess ocean  $CO_2$  sink is mainly chemically driven, but can additionally be altered by biological feedbacks. Further, net primary production decreases by up to 15% due to indirect effects of OAE. Our results do not confirm the ecological realization of the direct, physiologically positive effect of OAE on calcifying coccolithophores. Limiting alkalinity addition in locations with high aragonite saturation states is beneficial as it not only reduces the OAE impact on phytoplankton but also increases the reduction in atmospheric  $pCO_2$ . Our study highlights the need to take ecosystem responses into account when evaluating the effectiveness of OAE.

# 1 Introduction

In order to limit global warming to well below 2°C above preindustrial levels by 2100 as strived for in the Paris Agreement (UNFCCC, 2015), rapid phasing out of fossil fuels is required (Ho, 2023; Oschlies et al., 2023). However, to offset residual or hard-to-abate emissions such as carbon dioxide (CO<sub>2</sub>) emissions from aviation and maritime transport as well as non-CO<sub>2</sub> greenhouse gas emissions from agriculture (Oschlies et al., 2023; Ho, 2023), a portfolio of different carbon dioxide removal (CDR) technologies would require to sequester about 6–10 Gt atmospheric CO<sub>2</sub> per year (equivalent to 1.6–2.7 Pg C year<sup>-1</sup>, Smith et al., 2024). Indeed, all scenarios of the 6th report of the Intergovernmental Panel on Climate Change (IPCC) that limit warming to 1.5–2°C above preindustrial levels assume CDR implementation (Buck et al., 2023; Smith et al., 2024). Because terrestrial CDR technologies are often limited by competition for area (Fuss et al., 2014; Boysen et al., 2017; Friedlingstein et al., 2019), marine CDR technologies attract increasing attention (Oschlies et al., 2023; Doney et al., 2025). One of the most promising ocean-based approaches is ocean alkalinity enhancement (OAE, Köhler et al., 2013; Burns and Corbett, 2020; Gattuso et al., 2021).

The concept of OAE is to shift the ocean carbonate equilibria by adding alkaline substrates to the water. Simplified, total alkalinity quantifies carbonate and bicarbonate ion charges in the ocean (Zeebe and Wolf-Gladrow, 2001). Increasing alkalinity initially raises the *p*H of seawater, shifts the carbonate chemistry speciation towards lower aqueous CO<sub>2</sub> and higher carbonate ion concentration, and increases the saturation state of calcium carbonate (CaCO<sub>3</sub>). Ultimately, this allows additional atmospheric CO<sub>2</sub> to dissolve in seawater and be stored as bicarbonate or carbonate ions. Natural rock weathering, for example, stores about 0.3 Pg C year<sup>-1</sup> (IPCC, 2021). OAE efforts aim to mimic this natural process by actively deploying natural or artificially produced alkaline material to the surface ocean (Renforth and Henderson, 2017; Caserini et al., 2022). One of the possible alkalinity sources is calcium oxide or quicklime (CaO), which can be derived from limestone (a raw material for cement production) in a chemical process (Stanmore and Gilot, 2005; Caserini et al., 2022; Foteinis et al., 2022). In comparison to other alkaline material such as olivine and basalt, CaO shows rapid near-surface dissolution (Fakhraee et al., 2023), enabling atmospheric CO<sub>2</sub> uptake by the ocean, which would not occur if dissolution happened at depth.

35

The main limitations of OAE are limited feedstock supply, CO<sub>2</sub> emissions during the production of the OAE substrate (Fakhraee et al., 2023) and secondary precipitation of CaCO<sub>3</sub> that can remove more alkalinity than was added (Moras et al., 2022; Schulz et al., 2023). Nonetheless, numerous model studies imply that adding alkalinity can be an efficient CDR method (e.g., Hauck et al., 2016; Feng et al., 2017; Butenschön et al., 2021; Palmiéri and Yool, 2024). A major gap in the current models assessing the efficiency of OAE is, however, the disregard of feedbacks between OAE and the ocean planktonic ecosystem (Fennel et al., 2023).

The shift from CO<sub>2</sub> to bicarbonate and carbonate could potentially drive primary producers into CO<sub>2</sub> limitation, especially when the use of bicarbonate for photosynthesis is limited (Riebesell et al., 1993; Bach et al., 2019). In a review, Bach et al. (2019) argue that a transient shift in carbonate chemistry conditions should have little impact on the overall productivity. Indeed, there is little evidence for a harmful impact of enhanced alkalinity on the plankton community (Ramírez et al., 2024). Yet, changes in the phytoplankton species composition may be triggered, for example by a competitive advantage of small over large cells due to more efficient diffusion of CO<sub>2</sub> to the cell surface (Wolf-Gladrow and Riebesell, 1997; Chrachri et al., 2018; Bach et al., 2019). Bach et al. (2019) further hypothesize that calcifiers may become more important in regions of alkalinity deployment (change from "blue ocean" to "white ocean") due to carbonate chemistry conditions that are more favorable for calcification. In particular, this may hold under strong alkalinity enhancement, whereas weaker alkalinity addition could mainly counteract the negative impacts of ocean acidification on calcification (Lehmann and Bach, 2025). However, experimental studies show little to no effects of OAE on calcification (Subhas et al., 2022; Faucher et al., 2025; Bednaršek et al., 2025), especially as long as fluctuations in pH remain in a natural range (Gately et al., 2023), and partly even negative effects on the growth of calcifiers (Faucher et al., 2025). Furthermore, OAE was observed to have no (Gately et al., 2023) or a small negative effect (Ferderer et al., 2022) on silicification and has the potential to modify the carbon-to-nitrogen ratio (Ferderer et al., 2022). While the effects of OAE on calcifiers and silicifiers remain vague, Paul et al. (2024) show that increasing alkalinity can modify the nitrogen turnover, leading to a higher carbon-to-nitrogen ratio in particulate organic matter and ultimately to a decrease in the food quality.

60

In contrast to these uncertain effects of OAE on the ecosystem, calcification has a distinct effect on OAE. The formation of one mole CaCO<sub>3</sub> removes two moles of alkalinity - this so-called "leakage term" reduces the efficiency of OAE (Bach et al., 2019; Ho et al., 2003). For example, a modelling study shows that the addition of nutrients along with alkalinity results in a proliferation of calcifiers, which in turn decreases surface alkalinity and, hence, efficiency relative to a model simulation with the addition of alkalinity alone (Nagwekar et al., 2024). Thus, changes in calcification rates and the distribution of calcifiers caused by OAE should be considered in modelling studies when accounting for atmospheric CO<sub>2</sub> removal through OAE. For the same reasons, abiotic calcium carbonate precipitation, triggered at high aragonite saturation states in high-alkaline environments, can reduce OAE efficiency (Suitner et al., 2024), but its implication for real-ocean OAE is still unknown (e.g., Hartmann et al., 2023).

70

Adressing a gap in the current OAE modelling approaches, our study investigates the link between large-scale OAE and the ocean ecosystem. In particular, we use the Alfred Wegener Institute Earth System Model to link carbonate system changes to phytoplankton growth and calcification, and changes in calcification and calcite dissolution to the OAE efficiency. This allows an improved understanding of OAE effects in the living ocean.

#### 75 2 Methods

#### 2.1 AWI-ESM-1-REcoM

We computed emission-driven simulations with the Alfred Wegener Institute Earth System Model (AWI-ESM-1-REcoM). AWI-ESM is based on the AWI Climate Model (AWI-CM1, Semmler et al., 2020), but includes dynamic vegetation on land (Reick et al., 2021). AWI-ESM-1-REcoM further includes the representation of the carbon cycle between land, ocean, and atmosphere. Ocean and sea ice are represented by the Finite Element Sea Ice-Ocean Model FESOM (FESOM1.4, Danilov et al., 2004; Wang et al., 2014). Ocean biogeochemistry is described by the Regulated Ecosystem Model REcoM (REcoM2, Hauck et al., 2013; Schourup-Kristensen et al., 2014). The ocean and ocean biogeochemistry components are discretized on an unstructured mesh, allowing a variable grid resolution (12–147 km, mean 76 km, median 41 km). The carbonate system in REcoM is computed across the entire water column by the mocsy 2.0 routine (Orr and Epitalon, 2015; Seifert et al., 2022). The atmospheric component of AWI-ESM is represented by the spectral atmospheric model ECHAM (version 6.3, Stevens et al., 2013; Giorgetta et al., 2013). Land dynamics are modeled by the land surface model JSBACH version 3.20 including dynamic vegetation and the soil carbon model Yasso (Reick et al., 2021).

# 2.2 Modifications in REcoM for carbonate system effects on phytoplankton

The ocean biogeochemistry model REcoM describes the cycling of carbon, nitrogen, silicon, iron, and oxygen (Hauck et al., 2013). In the control version (without carbonate system effects, hereafter called "NO-CSE"), the ecosystem consists of two phytoplankton groups (small phytoplankton, diatoms), two zooplankton groups (generic zooplankton, polar macrozooplankton), and two detritus groups (slow and fast-sinking, Karakuş et al., 2021). Calcification is proportional to the gross photosynthesis rate of 2% of the small phytoplankton group with a fixed particulate inorganic to organic carbon ratio (PIC : POC) $_{ref} = 1$ . Calcite dissolution scales only with depth. Further, the 3D ocean carbonate system as well as the  $CO_2$  flux between atmosphere and ocean is computed by the mocsy 2.0 routines (Orr and Epitalon, 2015). A more detailed description of REcoM can be found in Appendix A.

Deviating from this control version, three major code changes of REcoM were used in this study (further on called "CSE") as described in Seifert et al. (2022). Firstly, coccolithophores as a new group of explicitly calcifying phytoplankton were added to the ecosystem, replacing the fixed share of 2% calcifiers in the small phytoplankton group. Calcification in CSE is not only a function of the gross photosynthesis rate, but also dependent on temperature and dissolved inorganic nitrogen (DIN) limitation resulting in a variable PIC:POC ratio. Secondly, the CSE version accommodates for direct effects of OAE on calcification as well as on gross photosynthesis of all phytoplankton groups. Thirdly, the carbonate ion concentration, and not depth, determines the calcite dissolution.

105

110

100

To account for direct effects of alkalinity enhancement on phytoplankton, the gross photosynthesis and calcification functions in REcoM were supplemented by a  $CO_2$  term  $f(CO_2)$  that scales between zero and three (i.e., maximal three-fold increase of gross photosynthesis and calcification) depending on changes in the carbonate system. The term was initially developed to describe responses to ocean acidification (Bach et al., 2015; Seifert et al., 2022), but the underlying carbonate system manipulations in the experiments also allow for the use under OAE-relevant conditions, as realized by Bach et al. (2019). As systematic assessments of OAE effects on phytoplankton growth and calcification are missing to date, we assumed that growth and calcification responses to changes in alkalinity can be described by the same function as responses to ocean acidification. Gross photosynthesis PS, which represents the increase in biomass over time without considering loss processes, is defined as:

115 
$$PS_i = f(T)_i \cdot f(PAR)_i \cdot f(N)_i \cdot f(CO_2)_i,$$
 (1)

where  $f(T)_i$ ,  $f(PAR)_i$ , and  $f(N)_i$  describe the effects of temperature, photosynthetically active radiation (PAR) and nutrient availability on the gross photosynthesis rate  $PS_i$  of the phytoplankton functional group i (more details in Text S2). Calcification *Calc* of coccolithophores (denoted by j) is defined as:

$$Calc_{i} = PS_{i} \cdot C_{i} \cdot (PIC : POC)_{ref} \cdot f(T)_{i,calc} \cdot f(N)_{i,calc} \cdot f(CO_{2})_{calc}, \tag{2}$$

Figure 1. The  $CO_2$  factor of phytoplankton net photosynthesis rate and calcification following Eq. 3. (A) displays variations in the factor with increasing alkalinity at surface pressure, a constant temperature of  $20^{\circ}$ C, salinity of 35, zero silicate and phosphate concentrations, and a DIC concentration of  $1950 \text{ mmol m}^{-3}$ . Contour plots show changes in the  $CO_2$  factor of diatoms with (B) varying temperature, (C) varying DIC concentrations, and (D) both varying temperature and DIC concentrations at a constant alkalinity concentration of  $2150 \text{ mmol m}^{-3}$ . Carbonate system parameters for the plots were assumed to not be equilibrated with the atmosphere, and were computed with PyCO2SYS version 1.8.3 (Humphreys et al., 2022). Note that the  $CO_2$  factor could reach much higher values under carbonate system conditions that are different from the example shown here, but was limited to three in our model.

where  $C_j$  is the biomass of coccolithophores and  $(PIC:POC)_{ref}$  a reference PIC:POC ratio of one. The temperature and DIN dependencies of calcification,  $f(T)_{j,calc}$  and  $f(N)_{j,calc}$ , follow Krumhardt et al. (2017). Calcification decreases linearly at temperatures below 10.6°C. The dependence on DIN limitation (modified from the original phosphate limitation, Krumhardt et al., 2017) is described by a modified Michaelis-Menten equation. Both terms are explained in more detail in Appendix B and C. The  $CO_2$  factor  $f(CO_2)$  in  $PS_i$  and  $Calc_j$  (Eq. 1 and 2) is defined as:

125 
$$f(CO_2)_{i \ or \ calc} = \frac{a_i \cdot [HCO_3^-]}{b_i + [HCO_3^-]} - \exp(-c_i \cdot [CO_{2(aq)}]) - d_i \cdot 10^{-pH}.$$
 (3)

The parameters a, b, c, and d (Table A1) were derived from curve fitting to experimental phytoplankton growth data (Seifert et al., 2022), and [HCO $_3$ <sup>-</sup>], [CO $_2(aq)$ ], and pH are the concentrations of bicarbonate, dissolved CO $_2$  and pH in the surrounding seawater. The CO $_2$  factor is zero at low alkalinity (or low HCO $_3$ <sup>-</sup> concentrations), plateaus at medium alkalinity, and decreases at high alkalinity (or low CO $_2(aq)$  concentrations) (Fig. 1A). Low temperatures (Fig. 1B) and low concentrations of dissolved inorganic carbon (DIC; Fig. 1C) narrow the window of maximum CO $_2$  factor values. In turn, it is high at high temperature and DIC concentrations (Fig. 1D).

**Table 1.** Summary of the model simulations (2040-2100), with alkalinity additions starting in 2040).

| Simulation name  | Carbonate system effects on phytoplankton? | Amount of alkalinity added in 2100 (Tmol year <sup>-1</sup> ) |      |       |  |  |
|------------------|--------------------------------------------|---------------------------------------------------------------|------|-------|--|--|
|                  |                                            | Europe                                                        | USA  | China |  |  |
| CSE-NO-OAE       | yes                                        | 0.0                                                           | 0.0  | 0.0   |  |  |
| CSE-OAE-low      | yes                                        | 14.8                                                          | 12.1 | 24.8  |  |  |
| CSE-OAE-high     | yes                                        | 29.7                                                          | 23.9 | 49.6  |  |  |
| CSE-OAE-high-lim | yes                                        | 29.7                                                          | 23.9 | 44.7  |  |  |
| NO-CSE-NO-OAE    | no                                         | 0.0                                                           | 0.0  | 0.0   |  |  |
| NO-CSE-OAE-low   | no                                         | 14.8                                                          | 12.1 | 24.8  |  |  |
| NO-CSE-OAE-high  | no                                         | 29.7                                                          | 23.9 | 49.6  |  |  |

## 2.3 Model simulations

Model simulations for this study were branched off after an initial concentration-driven spinup (piControl) of 1051 years with a constant atmospheric CO<sub>2</sub> concentration of 278 ppm, and a subsequent emission-driven spinup (esm-piControl) of 871 years.

Additional 200 years of esm-piControl spinup with both the NO-CSE and the CSE model version (section 2.2) were computed in parallel before starting the historical simulations (1850-2014; HIST-NO-CSE and HIST-CSE). For the subsequent future simulations, the SSP5-3.4-OS scenario was used. It follows the initial ramp-up of emissions equal to the SSP5-8.5 scenario (the unmitigated baseline scenario) from 2015 to 2039 before strong emission reduction from 2040 onwards, reaching zero emissions in 2070 and net negative emissions thereafter (O'Neill et al., 2016). Net negative emissions in this scenario are obtained by CDR methods other than OAE and additive to the OAE-caused atmospheric CO<sub>2</sub> reduction investigated in our study.

Starting in 2040 we added alkalinity with two different concentrations ("OAE-low", "OAE-high") in both model versions (section 2.4). We also computed simulations without alkalinity addition for each model version ("NO-OAE"). Hence, we ended up with six simulations for 2040-2100: NO-CSE-NO-OAE, NO-CSE-OAE-low, NO-CSE-OAE-high, CSE-NO-OAE, CSE-OAE-low, CSE-OAE-high (Table 1). Further, we computed one simulation which builds on the CSE-OAE-high simulation but in which no alkalinity was added to a grid cell when the saturation state of aragonite exceeded 10 to avoid conditions that favour abiotic calcium carbonate precipitation (CSE-OAE-high-lim; Table 1). We analysed differences between CSE and NO-CSE simulations as well as between OAE-high / OAE-low and NO-OAE simulations using independent two-sample t-tests (significance level:  $\alpha = 0.05$ ) either for the annual means of the entire time series 2040-2100 or for the annual means of the last 10 years of the simulation (2091-2100).

**Figure 2.** Summary of the alkalinity deployment. (A) Deployment regions along the European (green), Chinese (magenta), and the US EEZ (blue). (B) Amount of alkalinity added per m<sup>-2</sup> annually between 2040 and 2100 in each deployment region (solid: OAE-high, dotted: OAE-low).

# 2.4 OAE mask and alkalinity deployment

Alkalinity was added to the surface ocean in the exclusive economic zones (EEZ; up to 200 nautical miles away from the coastlines) of Europe, the US, and China (Fig. 2A). Subpolar regions (north of 67.5°N), the Baltic Sea east of 9.5°E as well as small marginal seas and remote parts of the EEZ (e.g., islands in the Pacific and Atlantic Ocean, Greenland) were excluded from the mask. The amount of alkalinity added in the OAE-high simulations scales according to the availability of CaO from cement production (Foteinis et al., 2022), with increasing annual cement production and, hence, CaO additions from 2040 to 2100 (Fig. 2B). In the OAE-low simulations half of this amount was added to the EEZ. Given the vast growth of the cement production in China over the past decades (S. Foteinis, personal communication), the Chinese EEZ starts with the highest alkalinity addition in the beginning of the deployment time and begins to saturate by the end of the century, while the amounts added to the European and the US EEZ progressively increase over time. In each grid cell of the model, the addition scales with the relative sea-ice cover which is assumed to prevent the distribution of alkalinity to the surface ocean. While the European and the Chinese EEZ are not affected by sea ice, the deployment in the US EEZ as reported here is 2.5±0.6% (2.7±1.3%) smaller in the first (last) decade of the deployment than in the initial deployment mask. In 2100, the amounts of alkalinity in the OAE-high (OAE-low) simulations are: Europe: 29.7 (14.9) Tmol year<sup>-1</sup>; USA: 23.9 (12.1) Tmol year<sup>-1</sup>; China: 49.6 (24.8) Tmol year<sup>-1</sup>.

#### 2.5 Analysis

The efficiency of OAE ( $\eta$ CO<sub>2</sub>) is computed from excess volume-integrated DIC and alkalinity in the OAE relative to the respective NO-OAE simulations (Renforth and Henderson, 2017) as:

$$\eta \text{CO}_2 = \frac{\Delta \text{DIC}}{\Delta \text{Alkalinity}}.$$
 (4)

The reduction of the  $CO_2$  partial pressure in seawater  $(pCO_{2(aq)})$  due to biological carbon drawdown,  $\Delta pCO_{2(aq,bio)}$ , is computed based on the biological sources and sinks to DIC  $(\Delta DIC_{bio})$  as well as the surface ocean carbonate system averaged over the upper 100 m of the water column (Hauck and Völker, 2015; Oziel et al., 2025) with:

$$\Delta p \text{CO}_{2(\text{aq,bio})} = \frac{\Delta \text{DIC}_{bio}}{\gamma_{\text{DIC}}} \cdot p \text{CO}_{2(\text{aq})}.$$
 (5)

The buffer factor  $\gamma_{\text{DIC}} = \frac{\text{DIC}}{\text{R}}$  (Egleston et al., 2010), with R being the Revelle factor (ratio of the relative change in  $p\text{CO}_{2(aq)}$ ) to the relative change in DIC, Zeebe and Wolf-Gladrow, 2001). Sources of DIC<sub>bio</sub> are phytoplankton and zooplankton respiration, CaCO<sub>3</sub> dissolution, and remineralization of dissolved organic carbon, while photosynthesis and calcification are sinks of DIC<sub>bio</sub>. Due to its strong seasonality,  $p\text{CO}_{2(aq,bio)}$  is calculated monthly and then averaged annually. Furthermore,  $\Delta p\text{CO}_{2(aq,bio)}$  allows to artificially combine the effects of carbonate system states (buffer factor and  $p\text{CO}_{2(aq)}$ ) and biological feedbacks of different simulations on the biological  $p\text{CO}_2$  drawdown. Both measures,  $\eta\text{CO}_2$  and  $\Delta p\text{CO}_{2(aq,biol)}$ , are computed both for the global ocean and in the three deployment regions as mean over 2091-2100.

# 3 Results

#### 3.1 OAE efficiency and modification by biological feedbacks

The reduction in atmospheric  $pCO_2$  in the OAE-low and OAE-high simulations relative to the NO-OAE simulations ranges from 2.9 to 7.8  $\mu$ atm (mean 2091-2100) and scales roughly with the amount of alkalinity added (Fig. 3A). Surface alkalinity (mean 2091-2100) in the OAE-high (OAE-low) simulation increases by  $104-105 \,\mathrm{mmol}\,\mathrm{m}^{-3}$  (53–54 mmol m<sup>-3</sup>) in the European EEZ,  $100-106 \,\mathrm{mmol}\,\mathrm{m}^{-3}$  (50–52 mmol m<sup>-3</sup>) in the US EEZ, and  $619-649 \,\mathrm{mmol}\,\mathrm{m}^{-3}$  (313–321 mmol m<sup>-3</sup>) in the Chinese EEZ (range given for CSE and NO-CSE simulations). Relative to the simulation without alkalinity addition (NO-OAE), this is an increase in surface alkalinity of 5–30% for OAE-high (2–15% for OAE-low). Globally,  $pCO_{2(aq)}$  in the upper  $100 \,\mathrm{m}$  mainly follows the trajectory of the prescribed  $CO_2$  emissions (increasing  $pCO_{2(aq)}$  by about  $60 \,\mu$ atm until 2060, decreasing  $pCO_{2(aq)}$  thereafter), and differences between the CSE and the NO-CSE simulations without alkalinity addition are largely caused by the model setup (Fig. A1A). In the deployment regions,  $pCO_{2(aq)}$  in the NO-OAE simulations is additionally modified by regional dynamics in the air-sea  $CO_2$  fluxes (Fig. A1B–D). Furthermore, by 2100 the simulations differ according to the amount of alkalinity added, with highest  $pCO_{2(aq)}$  values in the NO-OAE and lowest in the OAE-high simulations. Reductions in near-surface air temperatures and sea surface temperatures relative to the NO-OAE simulations are non-significant. This is probably due to both the relatively small reduction in atmospheric  $pCO_2$  that may not result in temperature changes that

Figure 3. OAE effects on atmosphere and ocean carbon. (A) Anomalies in atmospheric  $pCO_2$  (negative = reduction) and (B) anomalies in the cumulative air-sea  $CO_2$  flux (positive = into the ocean) in the OAE simulations relative to the NO-OAE simulations for the time period of alkalinity addition (2040-2100, annual means).

**Table 2.** Anomalies in atmospheric  $pCO_2$  as well as cumulative air-sea and air-land  $CO_2$  fluxes relative to the NO-OAE simulations. Negative sign for  $\Delta$ atm.  $pCO_2$  represents a decrease in atmospheric  $pCO_2$  concentration. Positive signs for the air-sea and air-land  $CO_2$  fluxes indicate an increasing sink or decreasing source. Stars indicate significant differences to the respective NO-OAE simulations.

|                  | $\Delta$ atm. $p$ CO $_2$ ( $\mu$ atm), mean 2091–2100 | $\Delta$ cum. air-sea $CO_2$ flux $(Pg C)$ , 2100 | $\Delta$ cum. air-land $CO_2$ flux (Pg C), 2100 |
|------------------|--------------------------------------------------------|---------------------------------------------------|-------------------------------------------------|
| CSE-OAE-low      | -2.9                                                   | +12.6*                                            | $-0.5^*$                                        |
| CSE-OAE-high     | -5.7                                                   | +26.0*                                            | $-8.4^{*}$                                      |
| NO-CSE-OAE-low   | -5.0                                                   | +11.4*                                            | +1.4                                            |
| NO-CSE-OAE-high  | -7.8*                                                  | +22.4*                                            | $-0.8^{*}$                                      |
| CSE-OAE-high-lim | -6.1                                                   | +23.8*                                            | $-6.8^*$                                        |

go beyond natural variability as well as the time lag in the Earth system response, in line with Jeltsch-Thömmes et al. (2024).

OAE causes a significant increase of the global cumulative air-sea CO<sub>2</sub> flux in all simulations, ranging between 11.4 and 26.0 Pg C by 2100 (Fig. 3B and A2, Table 2). About half of the OAE-induced CO<sub>2</sub> flux change occurs outside the deployment regions (Table 3). Including areas of 1000 km distance from the deployment region, resulting in areas that are 3–4 times the size of the original deployment regions (Fig. A3), covers 80–90% of the excess CO<sub>2</sub> uptake (Table A2). Especially in the US and the Chinese EEZ, a considerable share of excess ocean CO<sub>2</sub> uptake happens in the >1000 km surrounding area of the deployment region (Fig. 4A–C). Within the deployment regions, OAE is least efficient in the Chinese EEZ (0.34–0.45) and most efficient in the US EEZ (0.65–0.88; Table 3). Efficiency in the Chinese EEZ increases when considering the surrounding 1000 km (0.55–0.67, Fig. 4F, Table A2). Factors that can decrease efficiencies relative to theoretical values are feedbacks from the land carbon cycle, alkalinity losses by calcification, as well as the transport of alkalinity into deeper water parcels where CO<sub>2</sub> exchange with the atmosphere is impossible.

**Table 3.** Relative contribution of the deployment areas to anomalies in cumulative global air-sea CO<sub>2</sub> fluxes, and OAE efficiencies within the deployment regions. "Rest": global minus deployment regions; "Global": including deployment regions.

|                 |        |      | oution to and<br>2 flux (%), |      | OAE ef | OAE efficiency (ΔDIC / ΔAlkalinity),<br>mean 2091-2100 |       |        |  |
|-----------------|--------|------|------------------------------|------|--------|--------------------------------------------------------|-------|--------|--|
|                 | Europe | USA  | China                        | Rest | Europe | USA                                                    | China | Global |  |
| CSE-OAE-low     | 11.7   | 10.0 | 31.2                         | 47.2 | 0.75   | 0.69                                                   | 0.45  | 0.72   |  |
| CSE-OAE-high    | 11.1   | 9.9  | 24.0                         | 55.0 | 0.67   | 0.88                                                   | 0.36  | 0.73   |  |
| NO-CSE-OAE-low  | 13.0   | 10.2 | 33.5                         | 43.2 | 0.55   | 0.76                                                   | 0.42  | 0.66   |  |
| NO-CSE-OAE-high | 11.8   | 11.2 | 26.7                         | 50.2 | 0.63   | 0.65                                                   | 0.34  | 0.62   |  |

Figure 4.  $CO_2$  fluxes and efficiencies around the alkalinity deployment regions. (A, B, C) Cumulative excess ocean  $CO_2$  uptake in 2100 within the deployment regions (0 km distance) and including vicinities of increasing distance (100–1500 km). (D, E, F) The same for the efficiency  $\eta CO_2$ . Locations and sizes of the vicinity areas are displayed in Fig. A3.

When accounting for carbonate system effects on phytoplankton, the ocean takes up 11-16% more excess  $CO_2$  than without these effects (12.6 versus 11.4 Pg C and 26.0 versus 22.4 Pg C, respectively; Table 2), which is also reflected in higher efficiency values in almost all simulations (Table 3). However, the reduction in atmospheric  $pCO_2$  is smaller due to the weakened land  $CO_2$  uptake (Table 2) likely resulting from the different state of the climate system (radiative forcing and resulting effects on, e.g., temperature, precipitation, winds). The stronger  $CO_2$  sink in the CSE simulations could solely be driven by lower initial alkalinity and DIC concentrations compared to the NO-CSE simulations (Fig. A4) which chemically favor the uptake of atmospheric  $CO_2$ . To identify whether carbonate system effects on phytoplankton play an additional role in enhancing the ocean  $CO_2$  uptake in the CSE simulations, we computed  $\Delta pCO_{2(aq,bio)}$  (Eq. 5).

Figure 5. Schematic figure of the biological  $pCO_2$  drawdown summarizing the results presented in Table A3. The size of the boxes represents the weaker biological  $pCO_2$  drawdown resulting from OAE. Modifications of the biological  $pCO_2$  drawdown by carbonate system effects on phytoplankton (green) and by the state of the carbonate system ("C system states", blue) are represented by changes in the height of the boxes for each deployment region and globally. Positive numbers are an increase in the biological  $pCO_2$  drawdown, negative numbers a decrease. Stars indicate significant changes. Note that the schematics scale only qualitatively, not quantitatively.

The biological pCO<sub>2</sub> drawdown is a function of the strength of primary production and of the buffer capacity of seawater. Here, we combine results of different simulations to disentangle the roles of changing marine pelagic net primary production (NPP) in response to OAE and of different carbonate system states for the simulated ocean carbon uptake. The biological pCO<sub>2</sub> drawdown is consistently smaller in simulations with OAE than in those without, independent of whether carbonate system effects on phytoplankton growth are represented (for the CSE simulations: by 22% in the European and US EEZ, by 62% in the Chinese EEZ, and by 5% globally; p-value < 0.05, Fig. 5, Table A3). This is because OAE increases the buffer capacity and thus reduces the imprint of NPP on pCO<sub>2</sub> drawdown (Hauck and Völker, 2015). On top of that, two competing processes are responsible for differences between the CSE and NO-CSE simulations. Firstly, the CSE effects alter NPP and thus  $pCO_{2(aq,bio)}$ . Globally, this biological response in the CSE simulation leads to a decrease in the  $pCO_{2(aq,bio)}$  drawdown  $(-1.4 \,\mu \text{atm year}^{-1} \text{ averaged over } 2091-2100, \text{ Fig. 5})$  and, hence, to a weaker ocean CO<sub>2</sub> sink. Secondly, however, the different carbonate system state leads to a slightly larger biological pCO<sub>2</sub> drawdown (+0.1  $\mu$ atm year<sup>-1</sup> averaged over 2091–2100, Fig. 5). Note that both effects on the global biological  $pCO_2$  drawdown are not significant. Thus, the stronger global ocean  $CO_2$  sink in the CSE simulations must be fully driven by the state of the carbonate system, facilitated by a lower alkalinity-to-DIC ratio (1.118 in the CSE versus 1.123 in the NO-CSE simulation), a resulting lower buffer capacity and a resulting larger efficiency of OAE (Hinrichs et al., 2023), and not by biological feedbacks. Consistent with the findings in Hinrichs et al. (2023), we find that the baseline alkalinity and DIC concentrations are pivotal for the surface ocean pCO<sub>2</sub> reduction after alkalinity addition, which highlights the need for a careful assessment of the initial carbonate system states in OAE model studies. However, we observe that carbonate system effects on phytoplankton can indeed locally enhance the ocean CO<sub>2</sub> uptake following OAE, for example in the Chinese EEZ where the biological  $pCO_2$  drawdown is significantly increased by  $+1.0 \,\mu$ atm year<sup>-1</sup> (Fig. 5).

225

230

**Figure 6.** Anomalies of marine net primary production (NPP) in response to OAE in (A) with carbonate system effects on phytoplankton (CSE-OAE – CSE-NO-OAE) and (B) without carbonate system effects (NO-CSE-OAE – NO-CSE-NO-OAE) simulations, respectively, plotted against anomalies in surface alkalinity. NPP and surface alkalinity anomalies were plotted separately for each year and deployment region. Regression lines, R<sup>2</sup>, and p-values were computed together for all datapoints in one panel. Grey shadings represent the 95% confidence interval of the fitted lines.

# 3.2 OAE effects on biology

In the deployment regions of the simulations with carbonate system effects on phytoplankton, marine NPP is lower with OAE relative to simulations without OAE, with a significant difference especially in the Chinese EEZ (up to -15% in 2091-2100, Table A4). Less well pronounced anomalies can be seen on the global scale in the CSE simulations as well as in all NO-CSE simulations, where changes in marine NPP can only be caused by indirect OAE effects such as modifications of the radiative balance, winds, and mixed layer depth (Table A4). In accordance with this, annual NPP anomalies are negatively correlated with surface alkalinity anomalies in the simulations with carbonate system effects on phytoplankton (p<0.05, Fig. 6A) but not those without (p>0.05, Fig. 6B).

245

240

Decreasing NPP in the CSE-OAE simulations within the deployment areas is mainly driven by diatoms (significant negative correlation of diatom NPP with amount of alkalinity added, Fig. 7A) and slightly dampened by small phytoplankton (significant positive correlation of small phytoplankton NPP and amount of alkalinity added, Fig. 7B). Enhanced small phytoplankton NPP cannot fully balance the lower diatom NPP because of its smaller contribution to overall NPP (according to a community analysis in the simulations averaged over five years prior to the alkalinity deployment, 3% small phytoplankton versus 97% diatoms contribute to NPP in the Chinese EEZ, 40% versus 58% in the US EEZ). Nevertheless, within the bounds of the phytoplankton community described in our model, this indicates a community shift towards fewer large cells (diatoms) and more small cells (small phytoplankton) assuming unchanged grazing pressure on each group. Modifications in small phytoplankton and diatom NPP in the NO-CSE-OAE simulations are not correlated with the amount of added alkalinity (Fig. A5A,B).

Figure 7. Anomalies of marine net primary production (NPP) and concentrations of calcium carbonate (CaCO<sub>3</sub>) in response to OAE (CSE-OAE – CSE-NO-OAE) plotted against anomalies in surface alkalinity. (A) Diatom NPP, (B) small phytoplankton NPP, (C) coccolithophore NPP, and (D) CaCO<sub>3</sub> produced by coccolithophores. NPP, CaCO<sub>3</sub>, and surface alkalinity anomalies were plotted separately for each year and deployment region. Regression lines,  $R^2$ , and p-values were computed together for all datapoints in one panel. Grey shadings represent the 95% confidence interval of the fitted lines.

Both coccolithophore NPP and CaCO<sub>3</sub> of the CSE-OAE simulations decrease significantly with increasing alkalinity addition inside the deployment regions (Fig. 7C,D, Table A5). In the simulation without carbonate system effects, where calcification is performed by a fixed share of small phytoplankton, changes in CaCO<sub>3</sub> and the amount of added alkalinity are not correlated (Fig. A5C, Table A5). While the PIC:POC ratio in the NO-CSE simulations is defined to be constant, both coccolithophore PIC and POC can vary independently from each other in the CSE simulations. A significantly higher PIC:POC ratio in the European EEZ and globally in the OAE-high simulation with carbonate system effects on phytoplankton (+0.01, resulting in PIC:POC ratios of 1.18 and 1.14, respectively, Table A5) points towards more strongly calcifying coccolithophores, while a decreasing PIC:POC ratio in the Chinese EEZ (-0.05, resulting in a ratio of 1.10, Table A5) reflects lighter calcifying coccolithophores.

260

To assess whether the CO<sub>2</sub> factor is the primary driver of the negative correlation between OAE and NPP as well as CaCO<sub>3</sub> concentration, we examined changes in the factor over the time period of the OAE deployment. In some years before 2070, the CO<sub>2</sub> factor is smaller in the OAE compared to the NO-OAE simulations in the European and the US EEZ because of the small OAE signal caused by low alkalinity additions (Fig. 8A,B,D,E). At the latest from 2070 onwards, the factor is always higher in the CSE-OAE simulations relative to the CSE-NO-OAE simulations (Fig. 8). Hence, OAE is always beneficial for gross photosynthesis and calcification within the assumptions of our model, and strengthened CO<sub>2</sub> limitation does not play a role for marine NPP on a regional and global level under sustained OAE.

Instead, the inverse response of small phytoplankton and diatoms to OAE is likely caused by a stronger increase in the CO<sub>2</sub> factor of small phytoplankton, which alone would lead to a higher photosynthesis rate (Fig. 8A,B,C). Indeed, our parameter choice in  $f(CO_2)_i$  allows for a higher CO<sub>2</sub> factor for small phytoplankton in comparison to diatoms under increasing alkalinity concentrations (Fig. 1A), pointing towards a competitive advantage of small phytoplankton over diatoms. It was shown in Seifert et al. (2022) that small modifications in the CO<sub>2</sub> factors can trigger considerable shifts in the phytoplankton community, even by a phytoplankton group that is less represented in the respective region. Hence, the unequal increase rather than a decrease in the CO<sub>2</sub> factor are the likely causes for decreasing diatom and increasing small phytoplankton NPP. Decreasing diatom NPP can be further enhanced by indirect OAE effects that are caused by OAE-induced modifications in the atmospheric CO<sub>2</sub> which changes the radiative balance, winds, and, finally, the mixed layer depth and other drivers that have the potential to modify bottom-up and top-down effects on phytoplankton (similar to Nagwekar et al., 2025).

The stronger increase in the CO<sub>2</sub> factor of calcification relative to coccolithophore NPP (Fig. 8D–F) would suggest that the PIC:POC ratio should increase by 2100. We see this in the European EEZ and globally (Table A5), but not in the other EEZs. Similar to our interpretation of decreasing diatom NPP, we hypothesize that indirect OAE effects (e.g., on the radiative balance, winds, mixed layer depth) as well as the competition with the other phytoplankton groups dominate here, which is supported by the fact that both coccolithophore NPP and CaCO<sub>3</sub> concentrations decrease relative to the NO-OAE simulations despite the increasing CO<sub>2</sub> factor. Furthermore, coccolithophore biomass (not NPP) can be modified by changes in the grazing pressure, resulting in changes in the PIC:POC ratio that deviate from modifications in NPP.

#### 3.3 Effects of limited OAE

Our simulations show that OAE can decrease NPP significantly through the coupling of direct OAE responses (i.e., the CO<sub>2</sub> factor) to indirect feedbacks (e.g., competition, cascading effects on bottom-up and top-down drivers). Adding half of the amount of alkalinity (CSE-OAE-low versus CSE-OAE-high) effectively reduces the decline of total NPP in the Chinese EEZ by 60% (3.96 g C m<sup>-2</sup> year<sup>-1</sup> versus 9.18 g C m<sup>-2</sup> year<sup>-1</sup>), but the NPP decline in the European EEZ is four times higher than in the OAE-high simulation (2.48 g C m<sup>-2</sup> year<sup>-1</sup> versus 0.61 g C m<sup>-2</sup> year<sup>-1</sup>, Table A4). Moreover, the reduction in atmospheric pCO<sub>2</sub> by 2100 scales with the amount of alkalinity added, reaching only about 50% in the CSE-OAE-low simulation compared to the CSE-OAE-high simulation (2.9  $\mu$ atm versus 5.7  $\mu$ atm, Table 2). This poses the question of whether a more

**Figure 8.** Anomalies of the CO<sub>2</sub> factor in the deployment regions of the CSE-OAE simulations for (A, B, C) diatoms and small phytoplankton, and (D, E, F) coccolithophores and calcification relative to the CSE-NO-OAE simulations. The factor was computed offline from the surface carbonate system using Eq. 3. Note the different y-axes for the three regions.

targeted limitation of alkalinity addition could mitigate OAE impacts on phytoplankton while preserving CDR effectiveness.

With the motivation to avoid conditions in which abiotic CaCO<sub>3</sub> precipitation could happen, we complemented a CSE-OAE-high simulation in which no alkalinity was added to a grid cell when the saturation state of aragonite exceeded 10 (CSE-OAE-high-lim). This threshold is only exceeded in the Chinese EEZ, reducing the amount of added alkalinity by up to 4 mol m<sup>-2</sup> year<sup>-1</sup> (about 10%) compared to the CSE-OAE-high simulation and dampening the increase in surface alkalinity to 469 mmol m<sup>-3</sup> (compared to 649 mmol m<sup>-3</sup> in the CSE-OAE-high simulation). The significant NPP decrease in the Chinese EEZ of the CSE-OAE-high-lim simulation is dampened to 48% of the CSE-OAE-high simulation (4.75 g C m<sup>-2</sup> year<sup>-1</sup> versus 9.18 g C m<sup>-2</sup> year<sup>-1</sup>), similar to the CSE-OAE-low simulation. The significant NPP decline in CSE-OAE-high-lim in the European EEZ is still stronger than in the CSE-OAE-high simulation (2.01 g C m<sup>-2</sup> year<sup>-1</sup> versus 0.61 g C m<sup>-2</sup> year<sup>-1</sup>), but less strong than in the CSE-OAE-low simulation (2.48 g C m<sup>-2</sup> year<sup>-1</sup>, Table A4). NPP in the US EEZ decreases, however, more (up to 37%) than in both, CSE-OAE-low and CSE-OAE-high. The decrease of the PIC:POC ratio in the Chinese EEZ vanishes, while PIC:POC anomalies in the other EEZs and globally are comparable to the CSE-OAE-high simulation (Table A5). Whereas the OAE effects on the ecosystem with limited alkalinity addition are often smaller compared to the CSE-OAE-high simulation, it even increases the CDR effectiveness: atmospheric *p*CO<sub>2</sub> in the CSE-OAE-high-lim simulation is reduced by 7% more (6.1 μatm) than in the CSE-OAE-high simulation (5.7 μatm, Table 2).

## 4 Discussion

Marine CDR approaches are often less well perceived by the public than terrestrial methods (Cox et al., 2021). This is explicitly true for approaches in which material is released, such as OAE (Bertram and Merk, 2020), highlighting the need to develop a robust understanding of the risks and uncertainties of OAE. Our study aims to shed light on the large-scale interaction between OAE and the marine ecosystem.

## 4.1 Half of the excess CO<sub>2</sub> uptake occurs outside the deployment regions

With up to 1 Pg C year<sup>-1</sup> (in the CSE-OAE-high simulation, with a global alkalinity input of 96 Tmol year<sup>-1</sup>), OAE has the potential to store more than three times as much atmospheric CO<sub>2</sub> in the ocean than natural rock weathering (0.3 Pg C year<sup>-1</sup>, IPCC, 2021) and about 40–60% of the residual and hard-to-abate emissions (1.6–2.7 Pg C year<sup>-1</sup>, Smith et al., 2024). Approximately 50% of this excess CO<sub>2</sub> flux occurs outside the deployment regions, partly even in the periphery of 1500 km, likely depending on the prevailing surface ocean currents in the deployment region that transport the alkaline material away from its initial injection site. Water transport can also modify the time in which alkalinized waters are in contact with the atmosphere, thus allowing gas exchange. This reduced time for equilibration is likely what we observe in the US EEZ (Fig. 4E). A similar share of 50% was observed in the coastal OAE model study of Palmiéri and Yool (2024). These dynamics emphasize the need for large-scale "Monitoring, reporting, and verification" (MRV) processes (quantify the efficiency of CDR activities) as they require to track patches of artificially elevated alkalinity beyond the deployment region to fully assess the excess CO<sub>2</sub> taken up by the ocean (Ho et al., 2023). Further complicating MRV, our study shows that the enhanced ocean sink by OAE is in parts compensated, or even overcompensated, by a reduced land sink, in line with previous studies (e.g., Palmiéri and Yool, 2024; Jeltsch-Thömmes et al., 2024). This leads to a smaller reduction in atmospheric pCO<sub>2</sub> than would be expected from monitoring the air-sea CO<sub>2</sub> flux alone (this study, Schwinger et al., 2024).

## 4.2 Biological feedbacks modify the strength of ocean CO<sub>2</sub> uptake via OAE

Globally, carbonate system effects on phytoplankton are not the reason for higher air-sea CO<sub>2</sub> fluxes in the CSE compared to the NO-CSE simulation, but rather dissimilarities in initial surface alkalinity and DIC caused by differences in representations of the CaCO<sub>3</sub> cycle in the two model versions. In the European and Chinese EEZ, however, we indeed see that carbonate system effects on phytoplankton increase the potential of the ocean to take up atmospheric CO<sub>2</sub> relative to the NO-CSE simulations. Additional investigations are needed for the effect of biological feedbacks on long-term organic carbon storage: We see higher export efficiency (POC flux at 100 m / total NPP, Henson et al. (2012)) and transfer efficiency (POC flux at 1000 m / POC flux at 100 m, Passow and Carlson (2012)) in the Chinese EEZ of the CSE simulation (0.6% and 2.6%, respectively) compared to the NO-CSE simulation (0.5% and 2.4%, respectively), but lower export and transfer efficiencies in the European EEZ (CSE: 6.2% and 8.0%, respectively; NO-CSE: 6.6% and 8.2%, respectively). This suggest more efficient organic carbon export to depth in the Chinese EEZ due to the biological feedbacks, but increased surface remineralization and, hence, less effective deep organic carbon storage as a result of biological effects in the European EEZ. This has implications for MRV as it suggests

that biological feedbacks need to be taken into account when assessing the excess ocean  $CO_2$  sink and export of organic carbon to the deep ocean resulting from OAE.

#### 350 4.3 Indirect OAE effects lead to decreasing NPP

We found that OAE-induced anomalies in surface alkalinity correlate with a decrease in total NPP. As the CO<sub>2</sub> factor, the primary link between changes in the carbonate system and phytoplankton photosynthesis, does not imply reduced growth, other indirect OAE effects must diminish NPP. In fact, this is in line with the hypothesis of Bach et al. (2019) that shifts in the carbonate system by OAE are too small to trigger a significant effect on productivity. An imbalance in the change of the CO<sub>2</sub> factor between phytoplankton groups can, yet, result in a modified habitat competition in bottom-up and top-down factors at the expense of the dominant phytoplankton group. A similar finding for the CO<sub>2</sub> factor, termed as "cascading effects", was described by Seifert et al. (2022). Positive or neutral OAE effects on phytoplankton in laboratory studies must, thus, not necessarily result in positive or no effects of OAE on primary producers in the real ocean. Point-observations of the plankton community before, during, and after OAE applications would increase the understanding of direct and indirect ecosystem responses to OAE.

Our simulations do not confirm the ecological realization of the physiologically beneficial effect of OAE on coccolithophores which was hypothesized by Bach et al. (2019). NPP of coccolithophores decreases with progressing OAE, and PIC:POC decreases in the Chinese EEZ with the highest alkalinity deployment rates - both increasing coccolithophore NPP and PIC:POC ratios would have been a sign of the "white ocean" (Bach et al., 2019). Accordingly, our study could not confirm the reduction of surface alkalinity by enhanced calcification which would reduce the efficiency of OAE (part of the "leakage term", Ho et al., 2023). It has to be noted, however, that our alkalinity addition is only up to 10% of the addition required to trigger coccolithophore proliferation according to Lehmann and Bach (2025) (1.1 Pmol year<sup>-1</sup> versus a total of 0.1 Pmol year<sup>-1</sup> in 2100 in our simulations). The authors also point out that this global response may be overridden by other environmental factors on a regional or local scale, which is likely the case in our study as well. Furthermore, coccolithophores constitute only a very small part (<0.1–0.8%) of the phytoplankton community in the deployment regions of our model. In the real ocean, however, the deployment regions do host coccolithophore blooms (Daniels et al., 2018). Repeating the simulations with OAE deployment in coccolithophore hotspots of our model (e.g., the North Atlantic and the Equatorial Pacific) instead of the EEZs or improving the coastal coccolithophore representation in our model could provide further insights into the alkalinity leakage.

# 375 4.4 Maximizing effectiveness and minimizing environmental impacts requires delicate selection of OAE amount and location

Deploying only half of the alkalinity reduces atmospheric  $pCO_2$  by 50% as expected, but at the benefit of mitigating negative effects on the ecosystem. Hence, reducing the OAE deployment more locally (in this study, the reduction depends on the saturation state of aragonite) can be as or even more effective than deploying as much alkalinity as possible, while minimizing the effects on the ecosystem. In the real ocean this would also reduce the risk of secondary mineral precipitation, a process that

we do not parameterize in our model. Hence, in future OAE applications it is about finding the optimum between effectiveness and environmental impact. This optimum is likely unique for each deployment region, which hinders us to give quantitative suggestions for the ideal amount of alkalinity addition.

#### 4.5 Limitations of the study

We only consider alkalinity effects on phytoplankton, but zooplankton may be equally sensitive to OAE. Although the study of Sánchez et al. (2024) reveals that the plankton food-web in a mesocosm experiment was relatively resistant to OAE, the authors list potential vulnerabilities that may appear in other plankton communities. For example, zooplankton could be affected by OAE-induced changes in the nutritional value of their prey (Ferderer et al., 2022; Subhas et al., 2022; Bhaumik et al., 2025). Studying community-level effects of OAE in ecosystem models would reveal a better understanding of its large-scale ecosystem effects, but is currently impeded by lacking data on zooplankton-OAE interactions for model parameterizations.

Just as other Earth System Models (e.g., Hinrichs et al., 2023; Planchat et al., 2023), both the CSE and the NO-CSE version have a bias towards low surface alkalinity in comparison to observations (Fig. A4). We consider especially the representation of calcium carbonate dissolution above the saturation horizon as well as the improved biogeographical representation of plankton calcification other than coccolithophores as worthy of improvement. As shown by Hinrichs et al. (2023), biases in the surface alkalinity can indeed lead to an overestimation of the excess CO<sub>2</sub> uptake, which should be taken into account when transferring model findings to real ocean applications.

Ensemble simulations would help to quantify the effects of internal variability, thereby giving a better idea of the potential indirect effects of OAE on the ecosystem. However, since our study focuses on direct OAE effects in the upper ocean where the signal-to-noise ratio is high, we consider our study as a robust first step towards a better understanding of the mechanistic effects of OAE on phytoplankton.

#### 5 Conclusions

395

We show that biological feedbacks can modify the OAE efficiency, and that indirect OAE effects have the potential to alter phytoplankton community compositions. The physiologically beneficial effect of OAE on calcifying coccolithophores, as brought up in the "white ocean" hypothesis of Bach et al. (2019), is ecologically not realized in our simulations. Our results highlight the need to consider OAE-ecosystem feedbacks when investigating the effectiveness and the environmental impact of OAE. While experimental and mesocosm studies on OAE effects are increasing, little of these findings is used in models so far. Indeed, findings from laboratory and mesocosm experiments based on discrete samples can often not be directly used in models which are parameterized by continuous functions. Thanks to the large number of studies on phytoplankton responses to carbonate system changes, such parameterizations could be developed from data compilations (Bach et al., 2015; Seifert et al., 2022). However, for other potentially relevant OAE effects on phytoplankton such as responses in elemental ratios (e.g., Burkhardt

et al., 1999; Ferderer et al., 2022; Bhaumik et al., 2025), not to mention reactions of zooplankton to OAE, both the number of studies as well as the experimental designs are presently not sufficient to create model parameterizations. Ideally, model parameterizations are informed by numerous gradient-designed, single-species experiments using species that are representative for the plankton functional groups applied in models. Closer collaborations between experimental and modelling scientists can improve the projections of real-world OAE applications, and ultimately help to find a balance between environmental safety and OAE as a necessary CO2 removal technique to reduce climate change impacts.

Code and data availability. During the revision process, model data is available on figshare via a private link (Seifert et al., 2025). The data will be published with a doi link once the paper has been accepted. The REcoM code is the same as in Seifert et al. (2022) and available online (Seifert and Hauck, 2022).

# Appendix A: Description of REcoM

Our ocean biogeochemistry model REcoM is characterized by representing the ecosystem with variable intracellular stoichiometric ratios (carbon: nitrogen: chlorophyll for phytoplankton, Schartau et al., 2007; Hauck et al., 2013) which allow a flexible adaptation to prevailing environmental conditions following the photoacclimation model of Geider et al. (1998). The phytoplankton functional group of diatoms additionally incorporates a flexible stoichiometry for silicic acid, with varying degree of silicification through the decoupling between nutrient uptake and silicification (Claquin et al., 2002; Hohn, 2009). Intracellular iron concentrations are derived from a fixed iron: nitrogen ratio. While the classification of diatoms is taxonomic, the small phytoplankton comprises a wide range of taxa, such as non-silicifying and non-calcifying haptophytes and green algae.

Zooplankton intracellular stoichiometry is defined by carbon and nitrogen. The generic zooplankton group is distributed globally, while the polar macrozooplankton is restricted to the Southern Ocean and the northern high latitudes (Karakuş et al., 2021). Grazing is described by a sigmoidal function of variable preference (Fasham et al., 1990), with relatively higher grazing rates and a higher preference for small phytoplankton of the generic zooplankton, and relatively lower grazing rates and a higher preference for diatoms of the polar macrozooplankton (Seifert et al., 2022).

Sources for the slow-sinking detritus group are phytoplankton aggregation as well as sloppy feeding (which implicitly included defecation of the generic zooplankton group) and mortality of zooplankton. Fast-sinking detritus is only increased by sloppy feeding, mortality, and fecal pellet production of the polar macrozooplankton group. Detritus carbon and nitrogen is reduced by zooplankton grazing and the degradation to dissolved organic matter (Hauck et al., 2013; Karakuş et al., 2021). The sinking speed of the slow-sinking detritus groups is 20 m day<sup>-1</sup> at the surface and increases linearly with depth Hauck et al. (2013). The sinking speed of the fast-sinking detritus group is with 200 m day<sup>-1</sup> constant throughout the water column. Sinking

material that reaches the seafloor (single-layer sediment pool in REcoM) is subsequently released back to the lowermost depth layer of the water column.

# 445 Appendix B: Description of the phytoplankton gross photosynthesis in REcoM

Gross photosynthesis in REcoM of phytoplankton group i is dependent on temperature  $(f(T)_i)$ , PAR  $(f(PAR)_i)$ , the availability of nutrients  $(f(N)_i)$ , and - in the CSE simulations - the carbonate system  $(f(CO_2)_i)$ :

$$PS_i = f(T)_i \cdot f(PAR)_i \cdot f(N)_i \cdot f(CO_2)_i.$$
(B1)

The temperature dependence of diatom and small phytoplankton follows an Arrhenius equation:

$$450 \quad f(T)_i = PS_{max,i} \cdot \exp\left(-4500 \cdot \left[\frac{1}{T_K} - \frac{1}{T_{K,ref}}\right]\right),\tag{B2}$$

with  $T_K$  being the temperature in the water column in Kelvin, and  $T_{K,ref}$  being the reference temperature of 288.15 K (15°C).  $PS_{max,i}$  describes the group-specific maximum growth rate under non-limiting conditions at 15°C and is set to 3.5 d<sup>-1</sup> for diatoms and 3.0 d<sup>-1</sup> for small phytoplankton. For coccolithophores (denoted by a j), a different temperature dependence is used based on findings from experimental relations between coccolithophore growth rates and temperature (Fielding, 2013):

455 
$$f(T)_j = PS_{max,j} \cdot 0.1419 \cdot T_{\circ C}^{0.8151},$$
 (B3)

with  $T_{\circ C}$  being the temperature in the water column in degree Celsius and  $PS_{max,j}$  the scaling factor (2.8 d<sup>-1</sup>). To exclude coccolithophore growth in polar regions (see Seifert et al., 2022), the function was set to a small value (2.33 · 10<sup>-16</sup>) for temperatures below 0°C.

The dependence of gross photosynthesis on nutrient availability,  $f(N)_i$ , is determined by the most limiting nutrient, whereby limitation by DIN ( $l_i(DIN)$ ) and dissolved silicic acid ( $l_i(DSi)$ ) depend on the intracellular nitrogen or silicate-to-carbon ratios and the group-specific half-saturation constant for both nutrients (Hauck et al., 2013). Limitation by dissolved iron ( $l_i(DFe)$ ) is described by a Michaelis-Menten equation that depends on the group-specific half-saturation constant for DFe (for values of half-saturation constants see Seifert et al., 2022). The final nutrient limitation for coccolithophores and small phytoplankton is then described as:

$$f(N)_i = \min(l_i(DIN), l_i(DFe)), \tag{B4}$$

with 0 denoting complete limitation and 1 denoting no limitation. For diatoms, the limitation by DSi  $(l_i(DSi))$  is added within the minimum function.

The effect of PAR on gross photosynthesis,  $f(PAR)_i$ , follows Geider et al. (1998):

$$f(PAR)_i = 1 - \exp\left(\frac{-\alpha_j \cdot \mathbf{q}_j^{Chl:C} \cdot PAR}{f(T)_i \cdot f(N)_i}\right). \tag{B5}$$

In addition to the available PAR in the water column it depends on the group-specific maximum light harvesting efficiency  $\alpha_j$  (for numbers see Seifert et al., 2022), the variable chlorophyll-to-carbon ratio  $q_j^{Chl:C}$ , and the dependence of gross photosynthesis on temperature and nutrient availability, allowing for flexible adaptation to the prevailing light conditions depending on the available resources and temperature. The last term in the function of gross photosynthesis,  $f(CO_2)_i$ , is described in detail in the main text.

#### **Appendix C: Description of calcification in REcoM**

Calcification follows the description in Seifert et al. (2022), which builds on the model functions defined in Krumhardt et al. (2017). The temperature dependence of coccolithophore calcification (denoted by a j),  $f(T)_{j,calc}$ , is described as:

480 
$$f(T)_{j,calc} = \begin{cases} 0.104 \cdot \text{T}_{^{\circ}\text{C}} - 0.108 & \text{if T}_{^{\circ}\text{C}} 

**Figure A1.** Time series of average  $pCO_{2(aq)}$  in the upper 100 m of the watercolumn (A) globally and in the (B) European, (C) US, and (D) Chinese EEZ from the start of the alkalinity enhancement (2040) to the end of the century (2100).

**Figure A2.** Air-sea  $CO_2$  flux, mean of 2091-2100 (A, C, E) in the NO-CSE simulations, and (B, D, F) in the CSE simulations. Positive numbers in (A) and (B): ocean sink, negative numbers: ocean source. Fluxes in (C-F) are displayed as anomalies relative to the NO-OAE simulations; (C, D) for the OAE-low simulations, and (E, F) for the OAE-high simulations.

Figure A3. (A) Vicinities included in the computation of air-sea  $CO_2$  fluxes and efficiencies in addition to the deployment regions (dark red):  $+100 \,\mathrm{km}$  (lighter red),  $+200 \,\mathrm{km}$  (orange),  $+500 \,\mathrm{km}$  (yellow),  $+1000 \,\mathrm{km}$  (light blue),  $+1500 \,\mathrm{km}$  (dark blue). (B, C, D) Total area of the additional vicinities in the three deployment regions. Light grey lines indicate a linear increase from the area of the deployment regions to the area of the largest vicinity ( $+1500 \,\mathrm{km}$ ) for reference.

**Figure A4.** Anomalies in model surface ocean alkalinity compared to observations. (A) Surface ocean alkalinity in the gridded data product GLODAPv2 (data from 1971–2013; Lauvset et al., 2016). (B, C) Anomalies of the present-day surface alkalinity (mean 2010–2014 of the historical simulations minus GLODAPv2) in the CSE and the NO-CSE simulation.

**Figure A5.** Anomalies of marine net primary production (NPP) and concentrations of calcium carbonate (CaCO<sub>3</sub>) in the NO-CSE-OAE simulations relative to the NO-CSE-NO-OAE simulations plotted against anomalies in surface alkalinity. (A) Diatom NPP, (B) small phytoplankton NPP, and (C) CaCO<sub>3</sub> produced by a fixed share (2%) of small phytoplankton. NPP, CaCO<sub>3</sub>, and surface alkalinity anomalies were plotted separately for each year and deployment region. Regression lines, R<sup>2</sup>, and p-values were computed together for all datapoints in one panel. Grey shadings represent the 95% confidence interval of the fitted lines.

**Table A1.** Parameters for the  $CO_2$  factor (Eq. 3) following Seifert et al. (2022). PS = gross photosynthesis.

| Process                | a (dimensionless) | $b  (\mathrm{mol}  \mathrm{kg}^{-1})$ | $c  (\text{kg mol}^{-1})$ | $d  (\mathrm{kg}  \mathrm{mol}^{-1})$ |
|------------------------|-------------------|---------------------------------------|---------------------------|---------------------------------------|
| Diatom PS              | 1.040             | 28.90                                 | 0.8778                    | $2.640 \cdot 10^6$                    |
| Small phytoplankton PS | 1.162             | 48.88                                 | 0.255                     | $1.023\cdot 10^7$                     |
| Coccolithophore PS     | 1.109             | 37.67                                 | 0.3912                    | $9.450\cdot 10^6$                     |
| Calcification          | 1.102             | 42.38                                 | 0.7079                    | $1.343\cdot 10^7$                     |

**Table A2.** Relative contribution to anomalies in air-sea CO<sub>2</sub> fluxes and OAE efficiencies within the deployment regions and the 1000 km vicinity (vic.). "Rest": global minus deployment regions; "Global": including deployment regions.

|                 | 11010111         | • • • • • • • • • • • • • • • • • • • • | ition to an<br>flux (%), | ommer o | OAE efficiency ( $\Delta$ DIC / $\Delta$ Alkalinity), mean 2091 $-2100$ |               |                 |        |
|-----------------|------------------|-----------------------------------------|--------------------------|---------|-------------------------------------------------------------------------|---------------|-----------------|--------|
|                 | Europe<br>+ vic. | USA<br>+ vic.                           | China<br>+ vic.          | Rest    | Europe<br>+ vic.                                                        | USA<br>+ vic. | China<br>+ vic. | Global |
| CSE-OAE-low     | 14.3             | 20.6                                    | 51.7                     | 13.8    | 0.79                                                                    | 0.65          | 0.67            | 0.72   |
| CSE-OAE-high    | 13.8             | 21.4                                    | 43.1                     | 21.7    | 0.70                                                                    | 0.73          | 0.65            | 0.73   |
| NO-CSE-OAE-low  | 19.1             | 18.0                                    | 56.1                     | 6.8     | 0.57                                                                    | 0.57          | 0.55            | 0.66   |
| NO-CSE-OAE-high | 14.6             | 24.4                                    | 48.1                     | 12.9    | 0.65                                                                    | 0.66          | 0.55            | 0.62   |

Table A3. Annual sum of the surface ocean biological pCO<sub>2</sub> drawdown averaged over 2091-2100. Positive signs denote biological pCO<sub>2</sub> drawdown, negative signs denote a counteractive effect on biological  $pCO_2$  drawdown. Diff = OAE-high minus NO-OAE.

|                                                                               |                                                                                                          | Europe                | 8.         |                 | USA                           |            |       | China                         | a          |      | Global                | -          |      |
|-------------------------------------------------------------------------------|----------------------------------------------------------------------------------------------------------|-----------------------|------------|-----------------|-------------------------------|------------|-------|-------------------------------|------------|------|-----------------------|------------|------|
| Biological $pCO_2$ drawdown in the upper 100 m                                | Computation                                                                                              | $(\mu atm year^{-1})$ | u-1)       |                 | $(\mu \text{ atm year}^{-1})$ | 1)         |       | $(\mu \text{ atm year}^{-1})$ | r-1)       |      | $(\mu atm year^{-1})$ | r-1)       |      |
|                                                                               |                                                                                                          | OAE-<br>high          | NO-<br>OAE | Diff            | OAE-<br>high                  | NO-<br>OAE | Diff  | OAE-<br>high                  | NO-<br>OAE | Diff | OAE-<br>high          | NO-<br>OAE | Diff |
| with CO <sub>2</sub> effects on phytoplankton and earbonate system chances    | $\frac{\Delta \text{DIC}_{bio,CSE}}{\gamma \text{DIC}_{CSE}} \cdot p \text{CO}_{2(aq),CSE}$              | +16.4                 | +20.9      | s. <sub>4</sub> | +50.6                         | +65.0      | -14.4 | 44.9                          | +12.9      | -8.0 | +41.5                 | +43.6      | -2.1 |
| with $CO_2$ effects on phytoplankton and no carbonate system changes          | $\frac{\Delta \text{DIC}_{bio,CSE}}{\gamma \text{DIC}_{NO-CSE}} \cdot p^{\text{CO}_2(aq),NO-CSE} + 15.7$ | E +15.7               | +20.1      | 4<br>4.         | +47.6                         | +62.0      | -14.4 | 4.<br>8.                      | +11.9      | -7.1 | +38.6                 | +40.8      | -2.2 |
| with carbonate system changes but no $\mathrm{CO}_2$ effects on phytopiankton | $\frac{\Delta \text{DIC}_{bio,NO-CSE}}{\gamma \text{DIC}_{CSE}} \cdot p^{\text{CO}_2(aq),CSE}  ^{+19.9}$ | , +19.9               | +24.6      | t.<br>1.        | +55.1                         | +65.0      | -9.9  | +6.7                          | +15.7      | 0.6- | +41.0                 | +41.7      | -0.7 |
| Carbonate system effects                                                      | row 1 — row 2                                                                                            | +0.7                  | +0.8       | -0.1            | +3.0                          | +3.0       | 0.0   | +0.1                          | +1.0       | 6.0- | +2.9                  | +2.8       | +0.1 |
| Biological effects                                                            | row 1 - row 3                                                                                            | -3.5                  | -3.7       | +0.2            | 4.5                           | 0.0        | -4.5  | -1.8                          | -2.8       | +1.0 | +0.5                  | +1.9       | -1.4 |

**Table A4.** Anomalies of marine net primary production (NPP) in the OAE simulations relative to the NO-OAE simulations, averaged over 2091-2100. Stars indicate significant difference to the respective NO-OAE simulation.

|                  | Total NPP anomalies (g C m <sup><math>-2</math></sup> year <sup><math>-1</math></sup> ), mean 2091 $-2100$ |        |        |        | Relative NPP anomalies (%), mean 2091–2100 |       |        |        |
|------------------|------------------------------------------------------------------------------------------------------------|--------|--------|--------|--------------------------------------------|-------|--------|--------|
|                  | Europe                                                                                                     | USA    | China  | Global | Europe                                     | USA   | China  | Global |
| CSE-OAE-low      | -2.48*                                                                                                     | -2.24  | -3.96* | +0.10  | -4.2*                                      | -2.2  | -6.4*  | +0.2   |
| CSE-OAE-high     | -0.61                                                                                                      | -2.66* | -9.18* | -0.33  | -0.9                                       | -2.6* | -14.7* | -0.4   |
| NO-CSE-OAE-low   | -0.14                                                                                                      | +0.27  | -2.16* | +0.60* | -0.2                                       | +0.3  | -3.0*  | +0.8*  |
| NO-CSE-OAE-high  | -0.38                                                                                                      | -0.85  | -0.11  | +0.44* | -0.6                                       | -0.9  | -0.1   | +0.6   |
| CSE-OAE-high-lim | -2.01*                                                                                                     | -3.07* | -4.75* | -0.56  | -3.3*                                      | -3.1* | -7.6*  | -0.7   |

**Table A5.** Anomalies in phytoplankton particulate inorganic carbon (PIC) and particulate inorganic to organic carbon ratios (PIC:POC) in the OAE relative to the NO-OAE simulations. Because PIC:POC is set to a fixed ratio of 1 in the NO-CSE simulation, anomalies are denoted to be constant (const.). Stars indicate significant difference to the respective NO-OAE simulation.

|                  | Phytoplankton PIC anomalies $(10^{-3} \text{ Tg})$ , mean $2091-2100$ |        |       |         | Phytoplankton PIC:POC anomalies (mol C : mol C), mean 2091–2100 |        |        |        |  |
|------------------|-----------------------------------------------------------------------|--------|-------|---------|-----------------------------------------------------------------|--------|--------|--------|--|
|                  | Europe                                                                | USA    | China | Global  | Europe                                                          | USA    | China  | Global |  |
| CSE-OAE-low      | -34.6*                                                                | -79.8* | -0.3* | +105.5  | 0.00                                                            | 0.00   | -0.01  | 0.00   |  |
| CSE-OAE-high     | -16.6                                                                 | -99.9* | -0.4* | -337.0  | +0.01*                                                          | 0.00   | -0.05* | +0.01* |  |
| NO-CSE-OAE-low   | +12.2*                                                                | -8.7*  | -0.4  | +78.5   | const.                                                          | const. | const. | const. |  |
| NO-CSE-OAE-high  | +3.6                                                                  | +4.0   | +0.1  | +56.4   | const.                                                          | const. | const. | const. |  |
| CSE-OAE-high-lim | -17.7*                                                                | -94.0* | 0.0   | -633.2  | +0.01*                                                          | 0.0    | +0.01  | +0.01* |  |
|                  | Total phytoplankton PIC $(10^{-3} \text{ Tg})$ , mean $2091-2100$     |        |       |         | Phytoplankton PIC:POC (mol C: mol C), mean 2091–2100            |        |        |        |  |
| CSE-NO-OAE       | 94.7                                                                  | 253.4  | 1.2   | 19437.3 | 1.17                                                            | 1.06   | 1.15   | 1.13   |  |
| NO-CSE-NO-OAE    | 226.2                                                                 | 223.5  | 44.6  | 16811.9 | 1.00                                                            | 1.00   | 1.00   | 1.00   |  |

Author contributions. JH and CV aquired the funding for the study. MS, CV, and JH conceptualized the study. MS, CD, and JH developed the methodology, and MS and CD performed the model simulations. The data were investigated by MS with contributions from CD, CV, and JH, and visualizations were created by MS. MS prepared the manuscript with contributions from all co-authors.

Competing interests. The authors declare that they have no conflict of interest.

495

Acknowledgements. This study has received funding from the European Union's Horizon 2020 research and innovation programme under grant agreement No. 869357 (project OceanNETs, Ocean-based Negative Emission Technologies - analyzing the feasibility, risks, and cobenefits of ocean-based negative emission technologies for stabilizing the climate) and from the Initiative and Networking Fund of the Helmholtz Association (Helmholtz Young Investigator Group Marine Carbon and Ecosystem Feedbacks in the Earth System [MarESys], Grant VH-NG-1301). This study reflects only the author's view and the European Commission and their executive agency are not responsible for any use that may be made of the information it contains. The authors thank the colleagues in work package 4 of the OceanNETs project for fruitful and inspiring discussions. A special thank goes to Antti-Ilari Partanen and Tommi Bergman for the development of the OAE deployment mask. Furthermore, we thank Lennart Bach and Wentai Zhang for their very constructive reviews.

- Bach, L. T., Riebesell, U., Gutowska, M. A., Federwisch, L., and Schulz, K. G.: A unifying concept of coccolithophore sensitivity to changing carbonate chemistry embedded in an ecological framework, Progress in Oceanography, 135, 125–138, https://doi.org/10.1016/j.pocean.2015.04.012, 2015.
- Bach, L. T., Gill, S. J., Rickaby, R. E. M., Gore, S., and Renforth, P.: CO<sub>2</sub> Removal With Enhanced Weathering and Ocean Alkalinity Enhancement: Potential Risks and Co-benefits for Marine Pelagic Ecosystems, Frontiers in Climate, 1, https://doi.org/10.3389/fclim.2019.00007, 2019.
  - Bednaršek, N., van de Mortel, H., Pelletier, G., García-Reyes, M., Feely, R. A., and Dickson, A. G.: Assessment framework to predict sensitivity of marine calcifiers to ocean alkalinity enhancement identification of biological thresholds and importance of precautionary principle, Biogeosciences, 22, 473–498, https://doi.org/10.5194/bg-22-473-2025, 2025.
- Bertram, C. and Merk, C.: Public Perceptions of Ocean-Based Carbon Dioxide Removal: The Nature-Engineering Divide?, Frontiers in Climate, 2, https://doi.org/10.3389/fclim.2020.594194, 2020.
  - Bhaumik, A., Faucher, G., Henning, M., Meunier, C. L., and Boersma, M.: Prey dynamics as a buffer: enhancing copepod resilience to ocean alkalinity enhancement, Environmental Research Letters, 20, 024 058, https://doi.org/10.1088/1748-9326/adaa8c, 2025.
  - Boysen, L. R., Lucht, W., and Gerten, D.: Trade-offs for food production, nature conservation and climate limit the terrestrial carbon dioxide removal potential, Global Change Biology, 23, 4303–4317, https://doi.org/https://doi.org/10.1111/gcb.13745, 2017.
  - Buck, H. J., Carton, W., Lund, J. F., and Markusson, N.: Why residual emissions matter right now, Nature Climate Change, 13, 351–358, https://doi.org/10.1038/s41558-022-01592-2, 2023.
  - Burkhardt, S., Zondervan, I., and Riebesell, U.: Effect of CO<sub>2</sub> concentration on C:N:P ratio in marine phytoplankton: A species comparison, Limnology and Oceanography, 44, 683–690, https://doi.org/10.4319/lo.1999.44.3.0683, 1999.
- Burns, W. and Corbett, C. R.: Antacids for the Sea? Artificial Ocean Alkalinization and Climate Change, One Earth, 3, 154–156, https://doi.org/https://doi.org/10.1016/j.oneear.2020.07.016, 2020.
  - Butenschön, M., Lovato, T., Masina, S., Caserini, S., and Grosso, M.: Alkalinization Scenarios in the Mediterranean Sea for Efficient Removal of Atmospheric CO<sub>2</sub> and the Mitigation of Ocean Acidification, Frontiers in Climate, 3, 14, https://doi.org/10.3389/fclim.2021.614537, 2021.
- Caserini, S., Storni, N., and Grosso, M.: The Availability of Limestone and Other Raw Materials for Ocean Alkalinity Enhancement, Global Biogeochemical Cycles, 36, e2021GB007 246, https://doi.org/https://doi.org/10.1029/2021GB007246, 2022.
  - Chrachri, A., Hopkinson, B. M., Flynn, K., Brownlee, C., and Wheeler, G. L.: Dynamic changes in carbonate chemistry in the microenvironment around single marine phytoplankton cells, Nature Communications, 9, 74, https://doi.org/10.1038/s41467-017-02426-y, 2018.
- Claquin, P., Martin-Jézéquel, V., Kromkamp, J. C., Veldhuis, M. J. W., and Kraay, G. W.: Uncoupling of silicon compared with carbon and nitrogen metabolisms and the role of the cell cycle in continous cultures of *Thalassiosira pseudonana* (Bacillariophyceae) under light, nitrogen, and phosphorus control, Journal of Phycology, 38, 922–930, https://doi.org/https://doi.org/10.1046/j.1529-8817.2002.t01-1-01220.x, 2002.
  - Cox, E., Boettcher, M., Spence, E., and Bellamy, R.: Casting a Wider Net on Ocean NETs, Frontiers in Climate, 3, https://doi.org/10.3389/fclim.2021.576294, 2021.
- Daniels, C. J., Poulton, A. J., Balch, W. M., Marañón, E., Adey, T., Bowler, B. C., Cermeño, P., Charalampopoulou, A., Crawford, D. W., Drapeau, D., Feng, Y., Fernández, A., Fernández, E., Fragoso, G. M., González, N., Graziano, L. M., Heslop, R., Holligan, P. M., Hopkins,

- J., Huete-Ortega, M., Hutchins, D. A., Lam, P. J., Lipsen, M. S., López-Sandoval, D. C., Loucaides, S., Marchetti, A., Mayers, K. M. J., Rees, A. P., Sobrino, C., Tynan, E., and Tyrrell, T.: A global compilation of coccolithophore calcification rates, Earth System Science Data, 10, 1859–1876, https://doi.org/10.5194/essd-10-1859-2018, 2018.
- Danilov, S., Kivman, G., and Schröter, J.: A finite-element ocean model: principles and evaluation, Ocean Modelling, 6, 125–150, https://doi.org/https://doi.org/10.1016/S1463-5003(02)00063-X, 2004.
  - Doney, S., Lebling, K., Ashford, O., Pearce, C., Burns, W., Nawaz, S., Satterfield, T., Findlay, H., Gallo, N., Gattuso, J.-P., Halloran, P., Ho, D., Levin, L., Savoldelli, C., Singh, P., and Webb., R.: Principles for Responsible and Effective Marine Carbon Dioxide Removal Development and Governance, Washington, DC: WorldResources Institute, https://doi.org/10.69902/84b3a9a8, 2025.
- Egleston, E. S., Sabine, C. L., and Morel, F. M. M.: Revelle revisited: Buffer factors that quantify the response of ocean chemistry to changes in DIC and alkalinity, Global Biogeochem. Cycles, 24, https://doi.org/10.1029/2008GB003407, 2010.
  - Fakhraee, M., Li, Z., Planavsky, N. J., and Reinhard, C. T.: A biogeochemical model of mineral-based ocean alkalinity enhancement: impacts on the biological pump and ocean carbon uptake, Environmental Research Letters, 18, 044 047, https://doi.org/10.1088/1748-9326/acc9d4, 2023.
- Fasham, M. J. R., Ducklow, H. W., and McKelvie, S. M.: A nitrogen-based model of plankton dynamics in the oceanic mixed layer, Journal of Marine Research, 48, 591–639, https://doi.org/10.1357/002224090784984678, 1990.
  - Faucher, G., Haunost, M., Paul, A. J., Tietz, A. U. C., and Riebesell, U.: Growth response of *Emiliania huxleyi* to ocean alkalinity enhancement, Biogeosciences, 22, 405–415, https://doi.org/10.5194/bg-22-405-2025, 2025.
- Feng, E. Y., Koeve, W., Keller, D. P., and Oschlies, A.: Model-Based Assessment of the CO<sub>2</sub> Sequestration Potential of Coastal Ocean Alkalinization, Earth's Future, 5, 1252–1266, https://doi.org/https://doi.org/10.1002/2017EF000659, 2017.
  - Fennel, K., Long, M. C., Algar, C., Carter, B., Keller, D., Laurent, A., Mattern, J. P., Musgrave, R., Oschlies, A., Ostiguy, J., Palter, J. B., and Whitt, D. B.: Modelling considerations for research on ocean alkalinity enhancement (OAE), State of the Planet, 2-oae2023, 9, https://doi.org/10.5194/sp-2-oae2023-9-2023, 2023.
- Ferderer, A., Chase, Z., Kennedy, F., Schulz, K. G., and Bach, L. T.: Assessing the influence of ocean alkalinity enhancement on a coastal phytoplankton community, Biogeosciences, 19, 5375–5399, https://doi.org/10.5194/bg-19-5375-2022, 2022.
  - Fielding, S. R.: *Emiliania huxleyi* specific growth rate dependence on temperature, Limnology and Oceanography, 58, 663–666, https://doi.org/10.4319/lo.2013.58.2.0663, 2013.
  - Foteinis, S., Andresen, J., Campo, F., Caserini, S., and Renforth, P.: Life cycle assessment of ocean liming for carbon dioxide removal from the atmosphere, Journal of Cleaner Production, 370, 133 309, https://doi.org/https://doi.org/10.1016/j.jclepro.2022.133309, 2022.
- Friedlingstein, P., Allen, M., Canadell, J. G., Peters, G. P., and Seneviratne, S. I.: Comment on "The global tree restoration potential", Science, 366, eaay8060, https://doi.org/10.1126/science.aay8060, 2019.
  - Fuss, S., Canadell, J. G., Peters, G. P., Tavoni, M., Andrew, R. M., Ciais, P., Jackson, R. B., Jones, C. D., Kraxner, F., Nakicenovic, N., et al.: Betting on negative emissions, Nature climate change, 4, 850–853, https://doi.org/10.1038/nclimate2392, 2014.
  - Gately, J. A., Kim, S. M., Jin, B., Brzezinski, M. A., and Iglesias-Rodriguez, M. D.: Coccolithophores and diatoms resilient to ocean alkalinity enhancement: A glimpse of hope?, Science Advances, 9, eadg6066, https://doi.org/10.1126/sciadv.adg6066, 2023.

- Gattuso, J.-P., Williamson, P., Duarte, C. M., and Magnan, A. K.: The Potential for Ocean-Based Climate Action: Negative Emissions Technologies and Beyond, Frontiers in Climate, 2, https://doi.org/10.3389/fclim.2020.575716, 2021.
- Geider, R. J., MacIntyre, H. L., and Kana, T. M.: A dynamic regulatory model of phytoplanktonic acclimation to light, nutrients, and temperature, Limnology and Oceanography, 43, 679–694, https://doi.org/10.4319/lo.1998.43.4.0679, 1998.

- Giorgetta, M. A., Jungclaus, J., Reick, C. H., Legutke, S., Bader, J., Böttinger, M., Brovkin, V., Crueger, T., Esch, M., Fieg, K., Glushak, K., Gayler, V., Haak, H., Hollweg, H.-D., Ilyina, T., Kinne, S., Kornblueh, L., Matei, D., Mauritsen, T., Mikolajewicz, U., Mueller, W., Notz, D., Pithan, F., Raddatz, T., Rast, S., Redler, R., Roeckner, E., Schmidt, H., Schnur, R., Segschneider, J., Six, K. D., Stockhause, M., Timmreck, C., Wegner, J., Widmann, H., Wieners, K.-H., Claussen, M., Marotzke, J., and Stevens, B.: Climate and carbon cycle changes from 1850 to 2100 in MPI-ESM simulations for the Coupled Model Intercomparison Project phase 5, Journal of Advances in Modeling
   Earth Systems, 5, 572–597, https://doi.org/https://doi.org/10.1002/jame.20038, 2013.
  - Hartmann, J., Suitner, N., Lim, C., Schneider, J., Marín-Samper, L., Arístegui, J., Renforth, P., Taucher, J., and Riebesell, U.: Stability of alkalinity in ocean alkalinity enhancement (OAE) approaches consequences for durability of CO<sub>2</sub> storage, Biogeosciences, 20, 781–802, https://doi.org/10.5194/bg-20-781-2023, 2023.
- Hauck, J. and Völker, C.: Rising atmospheric CO<sub>2</sub> leads to large impact of biology on Southern Ocean CO<sub>2</sub> uptake via changes of the Revelle factor, Geophysical Research Letters, 42, 1459–1464, https://doi.org/10.1002/2015GL063070, 2015.
  - Hauck, J., Völker, C., Wang, T., Hoppema, M., Losch, M., and Wolf-Gladrow, D. A.: Seasonally different carbon flux changes in the Southern Ocean in response to the southern annular mode, Global Biogeochemical Cycles, 27, 1236–1245, https://doi.org/10.1002/2013GB004600, 2013.
  - Hauck, J., Köhler, P., Wolf-Gladrow, D., and Völker, C.: Iron fertilisation and century-scale effects of open ocean dissolution of olivine in a simulated CO<sub>2</sub> removal experiment, Environmental Research Letters, 11, 024 007, https://doi.org/10.1088/1748-9326/11/2/024007, 2016.

- Henson, S. A., Sanders, R., and Madsen, E.: Global patterns in efficiency of particulate organic carbon export and transfer to the deep ocean, Global Biogeochemical Cycles, 26, https://doi.org/https://doi.org/10.1029/2011GB004099, 2012.
- Hinrichs, C., Köhler, P., Völker, C., and Hauck, J.: Alkalinity biases in CMIP6 Earth system models and implications for simulated CO<sub>2</sub> drawdown via artificial alkalinity enhancement, Biogeosciences, 20, 3717–3735, https://doi.org/10.5194/bg-20-3717-2023, 2023.
- Ho, D.: Carbon dioxide removal is an ineffective time machine, Nature, 616, https://doi.org/https://doi.org/10.1038/d41586-023-00953-x, 2023.
  - Ho, D. T., Bopp, L., Palter, J. B., Long, M. C., Boyd, P. W., Neukermans, G., and Bach, L. T.: Monitoring, reporting, and verification for ocean alkalinity enhancement, State of the Planet, 2-oae2023, 12, https://doi.org/10.5194/sp-2-oae2023-12-2023, 2023.
  - Ho, T.-Y., Quigg, A., Finkel, Z. V., Milligan, A. J., Wyman, K., Falkowski, P. G., and Morel, F. M. M.: The elemental composition of some marine phytoplankton, Journal of Phycology, 39, 1145–1159, https://doi.org/10.1111/j.0022-3646.2003.03-090.x, 2003.
  - Hohn, S.: Coupling and decoupling of biogeochemical cycles in marine ecosystems, PhD thesis, Universität Bremen, http://nbn-resolving.de/urn:nbn:de:gbv:46-diss000112787, 2009.
  - Humphreys, M. P., Lewis, E. R., Sharp, J. D., and Pierrot, D.: PyCO2SYS v1.8: marine carbonate system calculations in Python, Geoscientific Model Development, 15, 15–43, https://doi.org/10.5194/gmd-15-15-2022, 2022.
- IPCC: Climate Change 2021: The Physical Science Basis. Contribution of Working Group I to the Sixth Assessment Report of the Intergovernmental Panel on Climate Change, V. Masson-Delmotte, P. Zhai, A. Pirani, S. L. Connors, C. Péan, S. Berger, N. Caud, Y. Chen, L. Goldfarb, M. I. Gomis, M. Huang, K. Leitzell, E. Lonnoy, J. B. R. Matthews, T. K. Maycock, T. Waterfield, O. Yelekçi, R. Yu, B. Zhou (eds), Cambridge University Press, 2021.
- Jeltsch-Thömmes, A., Tran, G., Lienert, S., Keller, D. P., Oschlies, A., and Joos, F.: Earth system responses to carbon dioxide removal as exemplified by ocean alkalinity enhancement: tradeoffs and lags, Environmental Research Letters, 19, 054 054, https://doi.org/10.1088/1748-9326/ad4401, 2024.

- Karakuş, O., Völker, C., Iversen, M., Hagen, W., Gladrow, D. W., Fach, B., and Hauck, J.: Modeling the impact of macrozooplankton on carbon export production in the Southern Ocean, Journal of Geophysical Research: Oceans, 126, e2021JC017315, https://doi.org/10.1029/2021JC017315, 2021.
- Köhler, P., Abrams, J. F., Völker, C., Hauck, J., and Wolf-Gladrow, D. A.: Geoengineering impact of open ocean dissolution of olivine on atmospheric CO<sub>2</sub>, surface ocean pH and marine biology, Environmental Research Letters, 8, 014 009, https://doi.org/10.1088/1748-9326/8/1/014009, 2013.
  - Krumhardt, K. M., Lovenduski, N. S., Iglesias-Rodriguez, M. D., and Kleypas, J. A.: Coccolithophore growth and calcification in a changing ocean, Progress in Oceanography, 159, 276–295, https://doi.org/10.1016/j.pocean.2017.10.007, 2017.
- Lauvset, S. K., Key, R. M., Olsen, A., van Heuven, S., Velo, A., Lin, X., Schirnick, C., Kozyr, A., Tanhua, T., Hoppema, M., Jutterström, S., Steinfeldt, R., Jeansson, E., Ishii, M., Perez, F. F., Suzuki, T., and Watelet, S.: A new global interior ocean mapped climatology: the 1° x 1° GLODAP version 2, Earth System Science Data, 8, 325–340, https://doi.org/10.5194/essd-8-325-2016, 2016.
  - Lehmann, N. and Bach, L. T.: Global carbonate chemistry gradients reveal a negative feedback on ocean alkalinity enhancement, Nature Geoscience, https://doi.org/10.1038/s41561-025-01644-0, 2025.
- Moras, C. A., Bach, L. T., Cyronak, T., Joannes-Boyau, R., and Schulz, K. G.: Ocean alkalinity enhancement avoiding runaway CaCO<sub>3</sub> precipitation during quick and hydrated lime dissolution, Biogeosciences, 19, 3537–3557, https://doi.org/10.5194/bg-19-3537-2022, 2022.
  - Nagwekar, T., Nissen, C., and Hauck, J.: Ocean Alkalinity Enhancement in Deep Water Formation Regions Under Low and High Emission Pathways, Earth's Future, 12, e2023EF004213, https://doi.org/https://doi.org/10.1029/2023EF004213, 2024.
- Nagwekar, T., Danek, C., Seifert, M., and Hauck, J.: Alkalinity Enhancement in the Subduction Regions: Efficiency, Earth System Feedbacks, and Deep Ocean Carbon Sequestration, submitted to: Environmental Research Letter, 2025.
  - O'Neill, B. C., Tebaldi, C., van Vuuren, D. P., Eyring, V., Friedlingstein, P., Hurtt, G., Knutti, R., Kriegler, E., Lamarque, J.-F., Lowe, J., Meehl, G. A., Moss, R., Riahi, K., and Sanderson, B. M.: The Scenario Model Intercomparison Project (ScenarioMIP) for CMIP6, Geoscientific Model Development, 9, 3461–3482, https://doi.org/10.5194/gmd-9-3461-2016, 2016.
  - Orr, J. and Epitalon, J.-M.: Improved routines to model the ocean carbonate system: Mocsy 2.0, Biogeosciences, 8, 485–499, https://doi.org/10.5194/gmd-8-485-2015, 2015.

- Oschlies, A., Bach, L. T., Rickaby, R. E. M., Satterfield, T., Webb, R., and Gattuso, J.-P.: Climate targets, carbon dioxide removal, and the potential role of ocean alkalinity enhancement, State of the Planet, 2-oae2023, 1, https://doi.org/10.5194/sp-2-oae2023-1-2023, 2023.
- Oziel, L., Gürses, O., Torres-Valdés, S., Hoppe, C. J. M., Rost, B., Karakuş, O., Danek, C., Koch, B. P., Nissen, C., Koldunov, N., Wang, Q., Völker, C., Iversen, M., Juhls, B., and Hauck, J.: Climate change and terrigenous inputs decrease the efficiency of the future Arctic Ocean's biological carbon pump, Nature Climate Change, https://doi.org/10.1038/s41558-024-02233-6, 2025.
- Palmiéri, J. and Yool, A.: Global-Scale Evaluation of Coastal Ocean Alkalinity Enhancement in a Fully Coupled Earth System Model, Earth's Future, 12, e2023EF004018, https://doi.org/10.1029/2023EF004018, 2024.
- Passow, U. and Carlson, C. A.: The biological pump in a high CO<sub>2</sub> world, Marine Ecology Progress Series, 470, 249–271, https://doi.org/10.3354/meps09985, 2012.
- Paul, A. J., Haunost, M., Goldenberg, S. U., Hartmann, J., Sánchez, N., Schneider, J., Suitner, N., and Riebesell, U.: Ocean alkalinity enhancement in an open ocean ecosystem: Biogeochemical responses and carbon storage durability, EGUsphere, 2024, 1–31, https://doi.org/10.5194/egusphere-2024-417, 2024.
  - Planchat, A., Kwiatkowski, L., Bopp, L., Torres, O., Christian, J. R., Butenschön, M., Lovato, T., Séférian, R., Chamberlain, M. A., Aumont, O., Watanabe, M., Yamamoto, A., Yool, A., Ilyina, T., Tsujino, H., Krumhardt, K. M., Schwinger, J., Tjiputra, J., Dunne, J. P., and Stock,

- C.: The representation of alkalinity and the carbonate pump from CMIP5 to CMIP6 Earth system models and implications for the carbon cycle, Biogeosciences, 20, 1195–1257, https://doi.org/10.5194/bg-20-1195-2023, 2023.
  - Ramírez, L., Pozzo-Pirotta, L. J., Trebec, A., Manzanares-Vázquez, V., Díez, J. L., Arístegui, J., Riebesell, U., Archer, S. D., and Segovia, M.: Ocean Alkalinity Enhancement (OAE) does not cause cellular stress in a phytoplankton community of the sub-tropical Atlantic Ocean, EGUsphere, 2024, 1–34, https://doi.org/10.5194/egusphere-2024-847, 2024.
- Reick, C. H., Gayler, V., Goll, D., Hagemann, S., Heidkamp, M., Nabel, J. E., Raddatz, T., Roeckner, E., Schnur, R., and Wilkenskjeld, S.: JSBACH 3-The land component of the MPI Earth System Model: documentation of version 3.2, 2021.
  - Renforth, P. and Henderson, G.: Assessing ocean alkalinity for carbon sequestration, Reviews of Geophysics, 55, 636–674, https://doi.org/10.1002/2016RG000533, 2017.
- Riebesell, U., Wolf-Gladrow, D. A., and Smetacek, V.: Carbon dioxide limitation of marine phytoplankton growth rates, Nature, 361, 249–660 251, 1993.
  - Schartau, M., Engel, A., Schröter, J., Thoms, S., Völker, C., and Wolf-Gladrow, D.: Modelling carbon overconsumption and the formation of extracellular particulate organic carbon, Biogeosciences, 4, 433–454, https://doi.org/10.5194/bg-4-433-2007, 2007.
  - Schourup-Kristensen, V., Sidorenko, D., Wolf-Gladrow, D. A., and Völker, C.: A skill assessment of the biogeochemical model REcoM2 coupled to the Finite Element Sea Ice–Ocean Model (FESOM 1.3), Geoscientific Model Development, 7, 2769–2802, https://doi.org/10.5194/gmd-7-2769-2014, 2014.
  - Schulz, K. G., Bach, L. T., and Dickson, A. G.: Seawater carbonate chemistry considerations for ocean alkalinity enhancement research: theory, measurements, and calculations, State of the Planet, 2-oae2023, 2, https://doi.org/10.5194/sp-2-oae2023-2-2023, 2023.
  - Schwinger, J., Bourgeois, T., and Rickels, W.: On the emission-path dependency of the efficiency of ocean alkalinity enhancement, Environmental Research Letters, 19, 074 067, https://doi.org/10.1088/1748-9326/ad5a27, 2024.
- Seifert, M. and Hauck, J.: REcoM code, Zenodo, https://zenodo.org/records/7457987, 2022.

680

- Seifert, M., Nissen, C., Rost, B., and Hauck, J.: Cascading effects augment the direct impact of CO<sub>2</sub> on phytoplankton growth in a biogeochemical model, Elementa: Science of the Anthropocene, 10, https://doi.org/10.1525/elementa.2021.00104, 00104, 2022.
- Seifert, M., Danek, C., Völker, C., and Hauck, J.: Model data for "Interactions between ocean alkalinity enhancement and phytoplankton in an Earth System Model", Figshare, https://figshare.com/s/f8dcd2ad0eb8b2c7b137, 2025.
- Semmler, T., Danilov, S., Gierz, P., Goessling, H. F., Hegewald, J., Hinrichs, C., Koldunov, N., Khosravi, N., Mu, L., Rackow, T., Sein, D. V., Sidorenko, D., Wang, Q., and Jung, T.: Simulations for CMIP6 with the AWI Climate Model AWI-CM-1-1, Journal of Advances in Modeling Earth Systems, 12, e2019MS002009, https://doi.org/10.1029/2019MS002009, 2020.
  - Smith, S. M., Geden, O., Gidden, M. J., Lamb, W. F., Nemet, G. F., Minx, J. C., Buck, H., Burke, J., Cox, E., Edwards, M. R., Fuss, S., Johnstone, I., Müller-Hansen, F., Pongratz, J., Probst, B. S., Roe, S., Schenuit, F., Schulte, I., and Vaughan, N. E.: The State of Carbon Dioxide Removal 2nd Edition, Tech. rep., https://doi.org/10.17605/OSF.IO/F85QJ, 2024.
  - Stanmore, B. R. and Gilot, P.: Review–calcination and carbonation of limestone during thermal cycling for CO<sub>2</sub> sequestration, Fuel Processing Technology, 86, 1707–1743, https://doi.org/10.1016/j.fuproc.2005.01.023, 2005.
  - Stevens, B., Giorgetta, M., Esch, M., Mauritsen, T., Crueger, T., Rast, S., Salzmann, M., Schmidt, H., Bader, J., Block, K., Brokopf, R., Fast, I., Kinne, S., Kornblueh, L., Lohmann, U., Pincus, R., Reichler, T., and Roeckner, E.: Atmospheric component of the MPI-M Earth System Model: ECHAM6, Journal of Advances in Modeling Earth Systems, 5, 146–172, https://doi.org/https://doi.org/10.1002/jame.20015, 2013.
- Subhas, A. V., Marx, L., Reynolds, S., Flohr, A., Mawji, E. W., Brown, P. J., and Cael, B. B.: Microbial ecosystem responses to alkalinity enhancement in the North Atlantic Subtropical Gyre, Frontiers in Climate, 4, https://doi.org/10.3389/fclim.2022.784997, 2022.

- Suitner, N., Faucher, G., Lim, C., Schneider, J., Moras, C. A., Riebesell, U., and Hartmann, J.: Ocean alkalinity enhancement approaches and the predictability of runaway precipitation processes: results of an experimental study to determine critical alkalinity ranges for safe and sustainable application scenarios, Biogeosciences, 21, 4587–4604, https://doi.org/10.5194/bg-21-4587-2024, 2024.
- Sánchez, N., Goldenberg, S. U., Brüggemann, D., Jaspers, C., Taucher, J., and Riebesell, U.: Plankton food web structure and productivity under ocean alkalinity enhancement, Science Advances, 10, eado0264, https://doi.org/10.1126/sciadv.ado0264, 2024.
- UNFCCC: United Nations Framework Convention on Climate Change: Paris Agreement, retrieved from https://unfccc.int/sites/default/files/english\_paris\_agreement.pdf, 08/01/2025, 2015.
- Wang, Q., Danilov, S., Sidorenko, D., Timmermann, R., Wekerle, C., Wang, X., Jung, T., and Schröter, J.: The Finite Element Sea Ice-Ocean Model (FESOM) v.1.4: formulation of an ocean general circulation model, Geoscientific Model Development, 7, 663–693, https://doi.org/10.5194/gmd-7-663-2014, 2014.
  - Wolf-Gladrow, D. and Riebesell, U.: Diffusion and reactions in the vicinity of plankton: A refined model for inorganic carbon transport, Marine Chemistry, 59, 17–34, https://doi.org/https://doi.org/10.1016/S0304-4203(97)00069-8, 1997.
- Zeebe, R. E. and Wolf-Gladrow, D.: CO<sub>2</sub> in seawater: Equilibrium, kinetics, isotopes, 65, Gulf Professional Publishing, 2001.