# Peer review of "Interactions between ocean alkalinity enhancement and phytoplankton in an Earth System Model"

_EGUsphere, 2025_

## Author Comment (AC1)

**Seifert et al. study the response of plankton communities to OAE in an Earth System Model. The study is very interesting and I enjoyed reviewing. I have some minor/moderate comments that may help the authors to further improve their study.**

We thank Lennart Bach for this positive evaluation and thoughtful evaluation that will help to improve our manuscript. We will carefully address the comments as listed below. Changes and additions to the text are highlighted in bold.

**Line 1: Have been investigated for what aspects? Efficiency?**

We will change this sentence to:

"**The functioning and efficiency of** ocean alkalinity enhancement (OAE) as a CO2 removal strategy …"

**Line 28: Consider double-checking this number. I think in IPCC 2013 the number is 0.25, which I think is based on global alkalinity discharge into the ocean via rivers as in Amiotte-Suchet et al 2003.**

We double-checked this number and couldn't find the estimate mentioned by the reviewer in the IPCC 2013 report. However, we found the number 0.3 Pg/yr in the IPCC 2013 report ("The Physical Science Base", Fig. 5.12). We will correct this number and reference in the text because no explanation for the lower number (0.14 Pg C yr-1, converted from 0.5 billion tons of CO2 per year) is provided in the reference that we cited initially (Renforth and Henderson 2017).

Accordingly, we will adapt the following sentence in the discussion (L. 319):

"With up to 1 Pg C year−1 (in the CSE-OAE-high simulation, with a global alkalinity input of 96 Tmol year−1), OAE has the potential to **store** more than **three** times as much atmospheric CO2 in the ocean than natural rock weathering (**0.3 Pg C year−1, IPCC, 2021**) and about 40−60% of the residual and hard-to-abate emissions (1.6−2.7 Pg C year−1, Smith et al., 2024)."

We also changed the term "sequester" to "store", as the definition of "sequestration" is currently under debate and likely refers to a permanent storage, whereby the time scale of carbon storage by OAE is highly dependent on factors such as ocean circulation and changes in atmospheric CO2.

**Line 37: I think there can still be some doubt on the efficiency of OAE, since models lack many of the relevant efficiency-modulating processes.**

As long as the effect on the ocean biogeochemistry dominates, which is especially the case for large amounts of alkalinity added as well as instantaneous surface dissolution, we would argue that ocean alkalinity enhancement is indeed an efficient CDR method. However, we agree that for local additions and/or small amounts of added alkalinity, the effectiveness of OAE can be questioned as efficiency-modulating processes (timing, hydrography, feedback reactions such

as secondary precipitation etc.) may diminish the CDR functioning of OAE. We will phrase this sentence more moderately:

"Nonetheless, **numerous model studies imply that adding alkalinity can be an efficient CDR method** (e.g., Hauck et al., 2016; Feng et al., 2017; Butenschön et al., 2021; Palmiéri and Yool, 2024)"

**Is equation 1 showing GPP as a biomass concentration change per time or a division rate? If a division rate then it would be somewhat different to what is generally considered GPP.**

This equation represents the increase of biomass over time without considering loss processes such as respiration, aggregation, grazing. Cell divisions are typically not represented in ocean biogeochemistry models which do not resolve single cells. We are aware that a change in total biomass is not necessarily linear to a change in cell number as cell size can be quite variable, too. However, while specific growth rates are commonly measured in laboratory studies, changes in phytoplankton carbon stocks are often not available. Hence, parameterizations in ocean biogeochemistry models must be based on the simplified assumption that the cell division rate is linear to the biomass change. We will add a note to the equation referring to the biomass change (L. 109):

"Gross photosynthesis, **which represents the increase in biomass over time without considering loss processes**, is defined as:"

**Line 119: "both parameterizations…" It took me quite a while to understand what "both" refers to. Consider providing a clearer description (some degree of repetition seems justifiable).**

We will change this to "The $CO_2$ factor $f(CO_2)$ in *PS* and *Calc* (Eq. 1 and 2) is defined as …."

**Fig. 1: Why can the CO2 factor range between 0 and 3 but Fig. 1a suggests ~1.2 is the maximum?**

Figure 1 presents the $CO_2$ factor for an example carbonate system (i.e., fixed pressure, temperature, salinity as well as silicate, phosphate, and DIC concentrations). In this case, the factor can only reach a value of about 1.2. However, in the model each of these fixed parameters is variable, and a different combination of parameter values could indeed lead to a factor that exceeds 1.2. To avoid a strong increase in the photosynthesis rate by a high $CO_2$ factor, we limit the factor to 3.0. We will add a sentence to the figure caption:

"**Note that the CO2 factor could reach much higher values under carbonate system conditions that are different from the example shown here, but was limited to three in our model.**"

**Fig. 1: Is the alkalinity on the x-axis "equilibrated" with atmospheric CO2 or "unequilibrated"? I would assume that stronger effects occur in the unequilibrated case. I think this needs to be clarified in the caption.**

The alkalinity is unequilibrated with atmospheric $CO_2$, i.e., $pCO_2$ is variable for each amount of alkalinity added. We will add this information to the caption:

"Carbonate system parameters for the plots **were assumed to not be equilibrated with the atmosphere, and** were computed with PyCO2SYS version …"

**Line 144: How can a t-test be done without any replicates? The statistical description probably needs more clarity.**

If we understand the question correctly, you assume that we are applying the t-test to two numbers only? We understand that one can get this impression when reading the methods / results. Actually, the t-test was applied to time series of annual means but reported in the context of the means of the time series. We will improve the method description to avoid confusion.

**The two paragraphs starting line 191 on OAE efficiency. The trends described here refer to where the air-sea CO2 influx occurred, whether in an EEZ of a deployment region or outside. I find this discussion somewhat artificial. Why does it matter where it occurs and how the efficiency is in a certain region? What matters is what happens over the entire affected area. The authors may consider deleting/condensing this section. These paragraphs also contain some discussion and not only results. Some readers may not like this (doesn't bother me but some really don't like interpretations in the results).**

We included this analysis to explain the low efficiencies in the Chinese EEZ, where actually the highest amounts of alkalinity were added. In our perspective it is important to mention that it is not enough to evaluate the efficiency within the region of alkalinity deployment, but that the transport of added alkalinity outside these regions and/or into other water layers need to be taken into account for MRV. We discuss this from L. 319 onwards. However, we agree with the reviewer that the analysis is a bit lengthy here and includes discussion parts. Furthermore, it overlaps with what we mention in the discussion (L. 319 and following). Hence, we will remove the part "Conversely, the efficiency …" (L. 201) to ".. larger than the actual deployment region" (L. 209).

For completeness we will add the following sentence to the discussion (L. 324):

"Approximately 50% of this excess CO2 flux occurs outside the deployment regions, partly even in the periphery of 1500 km, likely depending on the prevailing surface ocean currents in the deployment region that transport the alkaline material away from its initial injection site. **Water transport can also modify the time in which alkalinized waters are in contact with the atmosphere, thus allowing gas exchange. This reduced time for equilibration is likely what we observe in the US EEZ (Fig. 4E).** A similar share **of 50%** was observed in the coastal

OAE model study of Palmiéri and Yool (2024). These dynamics will complicate "Monitoring, reporting, and verification" (MRV) processes …"

**Table 2 (see above comment). Also, does the relatively high global efficiency of e.g. 0.72 include efficiency losses that would presumably come through OAE-induced reductions in the land-CO2 sink (or other Earth system feedbacks)? I would find this aspect more interesting than the text on air-sea CO2 flux.**

Exactly, OAE simulations in Earth System Models (with an interactive atmosphere as well as land feedbacks) generally indicate lower efficiencies than OAE simulations in ocean-only models (with prescribed atmosphere) because of the feedbacks from land. Other efficiency losses, for example by calcification and loss of the added alkalinity from the surface ocean prior to equilibrium, also exist in ocean-only models. We add this information to the text, and also refer to the higher efficiencies when talking about the CSE simulations.

We will add (L. 201):

"Efficiency in the Chinese EEZ increases when considering the surrounding 1000 km (0.55−0.67, Fig. 4F, Table A2). **Factors that can decrease efficiencies relative to theoretical values are feedbacks from the land carbon cycle, alkalinity losses by calcification, as well as the transport of alkalinity into deeper water parcels where CO2 exchange with the atmosphere is impossible.**"

And in L. 210:

"When accounting for carbonate system effects on phytoplankton, the ocean takes up 11−16% more excess CO2 than without these effects (Table 1)**, which is also reflected in higher efficiency values in almost all simulations (Table 2)**"

**Line 211: why is the land-sink weakened by an amount that overcompensates the phytoplankton effect? I would understand a mitigation of the land-sink but why is it relatively weaker than in the scenario without phytoplankton feedbacks? Adding an explanation would be interesting here.**

Modifications in the land sink are dependent on the sensitivities to the current concentration of CO2 in the atmosphere, but they also depend on changes that are triggered by the modified radiative forcing (temperature, winds etc). Hence, an overcompensation of the land sink, resulting in a relatively smaller reduction of atmospheric CO2 is possible in an earth system model. We will add to the sentence:

"However, the reduction in atmospheric pCO2 is smaller due to the weakened land CO2 uptake (Table 2) **likely resulting from the different state of the climate system (radiative forcing and resulting effects on, e.g., temperature, precipitation, winds).**"

**Line 212: Does that mean the differences in the run with the phytoplankton feedback is a spin-up artifact?**

The difference in alkalinity and DIC concentrations between the CSE and the NON-CSE versions is caused by the consideration of explicit calcification by coccolithophores in the CSE version. Thus, it is not a "spin-up artifact", as it is caused by differences in the model codes (implicit versus explicit representation of coccolithophores). We are aware that the model version with the explicit coccolithophore representation does not yet incorporate all components and processes that are relevant for a full representation of the carbonate pump (e.g., other plankton calcifiers such as foraminifera, other carbonate minerals such as aragonite, sufficient calcite dissolution above the saturation horizon). This may add to differences in surface alkalinity between the two model versions as well as between the model and observational data. Acknowledging this, the subsequent analysis (L. 216 and following) aimed to identify whether the difference in the excess CO2 uptake is only driven by the different initial conditions in the carbonate system, or whether biological effects play a role.

**Line 239: Does "globally" include land?**

NPP in our study only refers to marine pelagic NPP in the ocean and not terrestrial NPP. We will change this (also correcting the order in which the abbreviation is introduced) in L. 217 ("... to disentangle the roles of changing **marine pelagic net primary production (NPP)** in response to …"), in L. 237 ("… **marine NPP** is lower with OAE relative to …"), and in L. 239 ("... where changes in **marine** NPP are …" - sentence will be modified in response to the second reviewer). We will also change this in the captions of Figure 6, 7, and A5 and Table A4 ("Anomalies of **marine** net primary production (NPP) …").

**Fig. 5: "carbonate chemistry artifacts" sounds as if this is due to a glitch in the model. Was it an "effect" or really something weird in the model?**

The reviewer is right, "artifact" may not be the correct wording here. Indeed, it is an effect of changes in the calcification patterns that was caused by the explicit coccolithophore calcification (as opposed to the implicit calcification by a fixed share of small phytoplankton). We will change the wording in the figure to "**Change in biological pCO2 drawdown caused by different C system states**"

**Line 168: "…, rejection the hypothesis of strengthened CO2 limitation due to OAE." CO2 limitation can be seen in plankton community studies (usually a transient effect). So, I think the model cannot reject it. Rather you could say it plays no role for NPP on a regional/global level under stustained OAE?**

We agree, our original wording is too strong in this case. We will correct this to:

"Hence, OAE is always beneficial for gross photosynthesis and calcification within the assumptions of our model, **and strengthened CO2 limitation does not play a role for marine NPP on a regional and global level under sustained OAE.**"

**Line 271: It would be nice to have reminder here that a CO2 factor leads to higher growth (calcification).**

We will add:

"... is likely caused by a stronger increase in the CO2 factor of small phytoplankton, **which alone would lead to a higher photosynthesis rate** …"

**Line 284: Indirect effects would be MLD, light, temp, grazing etc.? Perhaps spell out.**

Yes, the reviewer is correct. We will make this sentence more explicit:

"Similar to our interpretation of decreasing diatom NPP, we hypothesize that indirect OAE effects **(e.g., on the radiative balance, winds, mixed layer depth)** as well as the competition with the other phytoplankton groups dominate here, which is supported by the fact that …"

**Line 290: The indirect effect response could be explained a bit better. It cannot be fully indirect because the origin must somehow be induced through the CSE-modifications in the model, right?**

Yes, the initial cause for changes in NPP must come from the CSE-modifications in the model. This is what we point to in this sentence (L. 290):
"Our simulations show that OAE can decrease NPP significantly through the coupling of direct OAE responses (i.e., the CO2 factor) to indirect feedbacks (e.g., competition, cascading effects on bottom-up and top-down drivers." The explanation was given in the previous subsection beginning in L. 271.

**Line 325: I am not sure if it really complicates things because it is already clear that constraining air-sea CO2 fluxes for individual deployments occurs at the basin scale and depends on large-scale (global) models. I think the discussion you are having here has already happened, and came to a conclusion.**

We will modify the sentence to:

"These dynamics **emphasize the need for large-scale** "Monitoring, reporting, and verification" (MRV) processes (quantify the effectiveness of CDR activities) as they require to track patches of artificially elevated alkalinity beyond the deployment region to fully assess the excess CO2 taken up by the ocean (Ho et al. 2023).

**Section 4.2: I think this is an interesting finding and needs further discussion. How does the increased NPP increase CO2 uptake? Nutrient concentrations are presumably the same between the runs, right? Hence, the shift in phytoplankton community composition somehow needs to export C more effectively to depth? Have you looked at preformed nutrient fields in the surface ocean? Are there indications of a more efficient BCP?**

The initial nutrient fields are the same, but the usage over time is different because of the different phytoplankton community compositions in the CSE and the NO-CSE model versions. However, we agree that increasing CO2 uptake only leads to enhanced long-term storage if the carbon is exported. Indeed, we see that export at 100m is smaller in the CSE compared to the

NO-CSE version, which may point towards more surface recycling and less carbon transport to depth. We will investigate this further and give a few numbers in the context of this discussion point.

**Line 351: Table A5 suggests that PIC:POC decreases in the Chinese EEZ.**

The reviewer is right, we made a mistake here. We will correct the sentence to:

"NPP of coccolithophores decreases with progressing OAE, and PIC:POC **decreases in the Chinese EEZ with the highest alkalinity deployment rates** - both increasing coccolithophore NPP and PIC:POC ratios would have been a sign of the "white ocean" (Bach et al., 2019)."

Furthermore, we will correct a typo in Table A5 ("PIC:POC" instead of "PIC:PIC").

**Line 350-352: I think the discussion non coccolithophore proliferation needs a slight adjustment: Instead of suggesting coccolithophores do not benefit I would rather say that they benefit, but relatively less than other groups in the model. You mention this above but I think it needs to be repeated here that a proliferation is physiologically possible but not ecologically realized. I think this also needs to be mentioned in the abstract as it is an important aspect of your study (and a very nice finding!).**

The reviewer has a good point here. According to the suggestion, we modify the first sentence in this paragraph to:
"Our simulations do not confirm the **ecological realization of the physiologically** beneficial effect of OAE on coccolithophores which was hypothesized by Bach et al. (2019)."

We will also change the corresponding sentence in the abstract to:

"Our results do not confirm the **ecological realization of the direct, physiologically positive** effect of OAE on calcifying coccolithophores."

**Lien 355: It is worth noting that the study by Lehmann and Bach also found that the correlation between PIC:POC and carbonate chemistry only holds when looking from "a very high altitude" (i.e. averaging over very large areas) but that regionally other factors override the CSYS response. I think this is at least conceptually a plausible explanation for differences, since under a limited perturbation the CSYS effect may be lost in noise (i.e. overridden by other factors that play a stronger role).**

Thank you for this note, we will add this information to the text by adding this sentence:

"It has to be noted, however, that our alkalinity addition is only up to 10% of the addition required to trigger coccolithophore proliferation according to Lehmann and Bach (2025) (1.1 Pmol year−1 versus a total of 0.1 Pmol year−1 in 2100 in our simulations). **The authors also point out that this global response may be overridden by other environmental factors on a regional or local scale, which is likely the case in our study as well.**"

**Line 363: You reduce by half and get 50% of CDR sounds not like a surprising outcome. I think you need to say what you would expect (in a one-dimensional assumption) to make clear why this is unexpected.**

The reviewer is correct, this sentence seems redundant, though it should be an opening sentence for the following discussion on the more regional OAE deployment. Hence, we will modify this sentence to:

"Deploying only half of the alkalinity **reduces atmospheric pCO2 reduction by 50% as expected, but at the benefit of mitigating negative effects on the ecosystem. Hence,** reducing the OAE deployment …"

**Section 4.4.: Did you have secondary precipitation in the model? I don't think so but the omega <10 discussion implies there is.**

No, we do not parameterize secondary precipitation; this information is given in the methods (L. 140) but we agree that it should be repeated in the discussion to avoid confusion. We will add the following sentence:

"... while minimizing the effects on the ecosystem. **In the real ocean this would also reduce the risk of secondary mineral precipitation, a process that we do not parameterize in our model.** Hence, in future …."

**Line 370: I don't think that the development for ocean acidification means that using the factor for OAE is a weakness. In the Bach et al study the mechanistic model was derived from treatments that are essentially "OAE", since alkalinity was manipulated in many different ways. They just didn't call it OAE back then, but this exact dataset was repurposed in the OAE context because it explored OAE-relevant conditions (Bach et al., 2019).**

Thank you for this first hand information! We will delete this entire paragraph and instead add a half-sentence to the methods:

"The term was initially developed to describe responses to ocean acidification (Bach et al., 2015; Seifert et al., 2022), **but the underlying carbonate system manipulations in the experiments also allow for the use under OAE-relevant conditions, as realized by Bach et al. 2019.**"

**The conclusion section could be improved. It currently just seems to open up an new discussion point (see next comment). Perhaps consider flashing out the key findings and novelties and then provide recommendations, what type of data exactly is needed to improve this model.**

We agree with the reviewer, the current conclusion is a bit superficial. We will revise this section, adding information on our key findings as well as on the data needed for models:

**"We show that biological feedbacks can modify the OAE efficiency, and that indirect OAE effects have the potential to alter phytoplankton community compositions. The physiologically beneficial effect of OAE on calcifying coccolithophores, as brought up in the "white ocean" hypothesis of Bach et al. (2019), is ecologically not realized in our simulations.** Our results highlight the need to consider OAE-ecosystem feedbacks when investigating the effectiveness and the environmental impact of OAE. While experimental and mesocosm studies on OAE effects are increasing, little of these findings is used **in models** so far. **Indeed, findings from laboratory and mesocosm experiments based on discrete samples can often not be directly used in models which are parameterized by continuous functions. Thanks to the large number of studies on phytoplankton responses to carbonate system changes, such parameterizations could be developed from data compilations (Bach et al., 2015; Seifert et al., 2022). However, for other potentially relevant OAE effects on phytoplankton such as responses in elemental ratios (e.g., Burkhardt et al., 1999; Ferderer et al., 2022; Bhaumik et al., 2025), not to mention reactions of zooplankton to OAE, both the number of studies as well as the experimental designs are presently not sufficient to create model parameterizations. Ideally, model parameterizations are informed by numerous gradient-designed, single-species experiments using species that are representative for the plankton functional groups applied in models.** Closer collaborations **between experimental and modelling scientists** can **improve the projections** of real-world OAE applications, and ultimately help to find a balance between environmental safety and OAE as a necessary CO2 removal technique to reduce climate change impacts."

We hereby focus on the experimental design. Details on other issues, such as differences in types and units of response variables in experiments and models, upscaling from single species or traits to plankton functional types etc. would be out of scope here, but should be a topic in future studies that aim to transfer knowledge from experiments to models.

**Line 392: In the context of closer collaboration: Can community studies really inform the development of these types of models? All you usually get from these studies is the ups and downs of some plankton groups, and the explanations WHY this happens are usually not particularly robust. It would be helpful to state what exactly the modellers need. My understanding is that models are driven by rates, and these come from physiological (or simplified ecological (grazing)) studies. Can you let the reader know what the big points for improvement are?**

"Community" in this context refers to us (scientists) and not to phytoplankton :-) We will change the wording from "used by the modelling community" to "**used in models**" to avoid confusion. Nevertheless, the reviewer is opening an interesting question - yes, models are based on rates / physiological understanding. On the other hand, they do not represent single species but so-called "plankton functional types". For example, instead of thermal performance curves (TPCs), models use exponential functions which represent envelope functions around the TPCs of a community. Hence, model parameterizations should ideally be based on several gradient designed, single species experiments, using species that are representative for the plankton

functional types. We will add this statement to our conclusion and give more details on the data needed to improve models regarding OAE effects on the ecosystem.

**Fig. A4: The figure implies that the model offset to GLODAP got worse upon implementation of the CSYS feedbacks. Does this need to be mentioned in the discussion or limitations section somewhere? Is that a concerning outcome, or just within the noise of general model competence?**

While the total difference between model and observations becomes slightly larger in the CSE simulation compared to the NO-CSE simulation, the error in the pattern / spatial variability is about the same. We agree that our model does not do the best job in representing surface alkalinity, just as many other Earth System Models (see, for example, Hinrichs et al. 2023, Planchat et al. 2023), and that effort is needed for a better representation of the carbonate pump and surface as well as deep ocean alkalinity. We add this to the limitation section of the study. However, we do not consider the slightly worse alkalinity representation in the CSE simulation as something concerning. We will add this paragraph to the limitation section:

"**Just as other Earth System Models (Hinrichs et al., 2023; Planchat et al,. 2023), both the CSE and the NO-CSE version have a bias towards low surface alkalinity in comparison to observations (Fig. A4). We consider especially the representation of calcium carbonate dissolution above the saturation horizon as well as the improved biogeographical representation of plankton calcification other than coccolithophores as worthy of improvement. As shown by Hinrichs et al., 2023, biases in the surface alkalinity can indeed lead to an overestimation of the excess CO2 uptake, which should be taken into account when transferring model findings to real ocean applications.**"

---

## Author Comment (AC2)

**General summary:**

**The authors investigate the potential impacts of Ocean Alkalinity Enhancement (OAE) on the Earth system by incorporating carbonate chemistry dependencies into the phytoplankton growth term within an Earth system model (ESM). They perform multiple ESM simulations to assess the environmental impacts of OAE, with particular emphasis on ecosystem responses. This approach highlights the importance of accounting for biological feedback when evaluating geoengineering strategies such as OAE.**

**This is a timely and important contribution to the fields of climate change and geoengineering. I recommend publication of this work after the authors address the concerns outlined below.**

We thank Wentai Zhang for this positive feedback and the constructive comments which will help us to improve our manuscript. We will carefully address each of the comments. Changes and additions to the text are highlighted in bold.

**Comments:**

**Line 20: Please add a reference to support the statement "Efforts…"**

We will rephrase this sentence to not give more weight to ocean-based than to land-based approaches, and add two references to highlight the increasing across-sector attention towards marine CDR technologies (the best practice guide for ocean alkalinity enhancement, Oschlies et al., 2023, and a recently published blue paper on principles for responsible and effective CDR development and governance, Doney et al. 2025):

"Because terrestrial CDR technologies are often limited by competition for area (Fuss et al., 2014; Boysen et al., 2017; Friedlingstein et al., 2019), **marine CDR technologies attract increasing attention (Oschlies et al., 2023; Doney et al., 2025)**."

**Line 40: I suggest deleting the phrase ", although it was identified as a major risk" to improve clarity and conciseness.**

Although OAE is mentioned as a potential ecological risk in Fennel et al. (2023) which is cited in the relevant sentence, this is not a main outcome of their study and we will delete the phrase as suggested by the reviewer.

**Line 45: Consider changing the word "minimal" to "little".**

We will change the wording as suggested by the reviewer.

**Line 62: The sentence beginning with "In a modeling study, …" is unclear. Please revise to clarify it.**

We agree that the original sentence was very nested. We will rephrase it to:

"**For example, a modelling study shows that the addition of nutrients along with alkalinity results in a proliferation of calcifiers, which in turn decreases surface alkalinity and, hence, efficiency relative to a model simulation with the addition of alkalinity alone (Nagwekar et al., 2024).**"

**Line 70: Please specify which Earth system model was used in this study.**

We will rephrase the sentence to:

"In particular, we use **the Alfred Wegener Institute Earth System Model to link** carbonate system changes to phytoplankton growth and calcification, and changes in calcification and calcite dissolution to the OAE efficiency."

**Line 210: I was unable to locate the 16% in Table 1. Please clarify where this number comes from.**

We computed the relative difference of the cumulative air-sea $CO_2$ fluxes between the simulations that are listed in Table 1. For clarification we will add these numbers to the sentence:

"… When accounting for carbonate system effects on phytoplankton, the ocean takes up 11−16% more excess $CO_2$ than without these effects (**12.6 versus 11.4 Pg C and 26.0 versus 22.4 Pg C, respectively;** Table 1) …"

**Line 221: Instead of ranges or general terms, please provide the exact values here.**

We will add the exact numbers for the CSE simulations to the text:

"The biological $pCO_2$ drawdown is consistently smaller in simulations with OAE than in those without, independent of whether carbonate system effects on phytoplankton growth are represented (**for the CSE simulations: by 22% in the European and US EEZ, by 62% in the Chinese EEZ, and by 5% globally**; p-value<0.05, Fig. 5, Table A3)."

**Line 239: The sentence "NPP anomalies globally…" is unclear. Please rephrase.**

We agree, the original sentence was not very clear. We will rephrase to:

"**Less pronounced anomalies can be seen on the global scale in the CSE simulations as well as in all NO-CSE simulations, where changes in marine NPP can only be caused by indirect OAE effects such as modifications of the radiative balance, winds, and mixed layer depth.**"

**Line 247: The origin of the values "3% vs 97%, 40% vs 58%" is unclear. Please indicate their source or explain how they were derived.**

The reviewer is right, these numbers are not indicated in any of the tables in the manuscript as they were derived from a community composition analysis averaged over the last five years prior to the start of the alkalinity deployment. We will add this information to the sentence:

"Enhanced small phytoplankton NPP cannot fully balance the lower diatom NPP because of its smaller contribution to overall NPP (**according to a community analysis in the simulations averaged over five years prior to the alkalinity deployment,** 3% small phytoplankton versus 97% diatoms **contribute to NPP** in the Chinese EEZ, and 40% versus 58% in the US EEZ)"

**Line 301: This sentence should be rewritten for clarify. Also, please specify where the associated values can be found in the text or tables.**

We will rephrase this sentence to:

"With the motivation to avoid conditions in which abiotic CaCO3 precipitation could happen, we complemented a CSE-OAE-high simulation in which no alkalinity was added to a grid cell when the saturation state of aragonite exceeded 10 (CSE-OAE-high-lim). **This threshold is only exceeded in the Chinese EEZ, reducing the amount of added alkalinity by up to 4 mol m−2 year−1 (about 10%) compared to the CSE-OAE-high simulation and dampening the increase in surface alkalinity to 469 mmol m−3 (compared to 649 mmol m−3 in the CSE-OAE-high simulation).**"

However, as these are the only numbers referring to the amount of added alkalinity as well as surface alkalinity in the CSE-OAE-high-lim simulations, we refrain from adding an extra table and prefer to only mention these numbers in the text.

**Figure A4B and A4C: These figures show the difference between modeled alkalinity and observational data. Please revise caption accordingly.**

The reviewer is absolutely right, the caption is incorrect. We will revise the caption.

**Section 2.3: It would be helpful to include a table summarizing the key details of all simulations conducted in this study.**

We will add a summary table for the model simulations to the manuscript.

**Figure 2,3, and 4: I recommend revising the captions to first provide an overview that describes the figure as a whole, followed by brief description for each subfigure individually. This will help readers better understand both the overall context and the specific content shown in each panel.**

We agree and will add the following overall sentences:

Fig. 2: "**Summary of the alkalinity deployment**."
Fig. 3: "**OAE effects on atmosphere and ocean carbon.**"
Fig. 4: "**CO2 fluxes and efficiencies around the alkalinity deployment regions.**"

**Please improve the abstract and conclusion section.**

We will modify the abstract according to the suggestions of the other reviewer. The conclusion section will be revised entirely to include our key findings as well as details on the type of data needed to improve the parameterization of OAE effects on biology in ocean biogeochemistry models.